# Cloud properties and their projected changes in CMIP models with low/medium/high climate sensitivity

Lisa Bock[1] and Axel Lauer[1]

[1]Deutsches Zentrum für Luft- und Raumfahrt (DLR), Institut für Physik der Atmosphäre, Oberpfaffenhofen, Germany

**Correspondence:** L. Bock (lisa.bock@dlr.de)

**Abstract.** Since the release of the first CMIP6 simulations one of the most discussed topics is the higher effective climate sensitivity (ECS) of some of the models resulting in an increased range of ECS values in CMIP6 compared to previous CMIP phases. An important contribution to ECS is the cloud climate feedback. Although climate models have continuously been developed and improved over the last decades, a realistic representation of clouds remains challenging. Clouds contribute to the large uncertainties in modeled ECS, as projected changes in cloud properties and cloud feedbacks also depend on the simulated present-day fields.

In this study we investigate the representation of both cloud physical and radiative properties from a total of 51 CMIP5 and CMIP6 models. ECS is used as a simple metric to group the models as the sensitivity of the physical cloud properties to warming is closely related to cloud feedbacks, which in turn are known to have a large contribution to ECS. Projected changes of cloud properties in future scenario simulations are analyzed by ECS group. In order to help interpreting the projected changes, model results from historical simulations are also compared to observations.

The results show that models in the high ECS group are typically in better agreement with satellite observations for total cloud cover and ice water path in midlatitudes, especially over the Southern Ocean. Notoriously difficult tasks, however, such as simulating clouds in the Tropics or the correct representation of stratocumulus clouds remain similarly challenging for all three ECS groups. Differences in the net cloud radiative effect as a reaction to warming and thus differences in effective climate sensitivity among the three ECS groups are found to be driven by changes in a range of cloud regimes rather than individual regions. In polar regions, high ECS models show a weaker increase in the net cooling effect of clouds due to warming than the low ECS models. At the same time, high ECS models show a decrease in the net cooling effect of clouds over the tropical ocean and the subtropical stratocumulus regions whereas low ECS models show either little change or even an increase in the cooling effect. Over the Southern Ocean, the low ECS models show a higher sensitivity of the net cloud radiative effect to warming than the high ECS models.

## 1  Introduction

Climate models are an essential tool for projecting future climate. Within the context of the Coupled Model Intercomparison Project (CMIP, https://www.wcrp-climate.org/wgcm-cmip), a World Climate Research Programme (WCRP) initiative, several modeling groups worldwide provide a set of coordinated simulations with different Earth system models (ESMs) of the past

(historical) time period and different future scenarios. The main objective of CMIP is to better understand past, present, and future climate, its variability and future change arising from both natural, unforced variability and in response to changes in radiative forcing in a multi-model context.

Across the different CMIP phases, several improvements e.g. in the climatological large-scale patterns of temperature, water vapor, and zonal wind speed were found with the latest phase models (CMIP6, Eyring et al. (2016)) typically performing slightly better than their CMIP3 and CMIP5 predecessors when compared to observations (Bock et al., 2020). While this is also the case for some cloud properties and selected regions such as the Southern Ocean, clouds remain challenging for global climate models with many known biases remaining in CMIP6 (Lauer et al., 2023). As such, clouds continue to play a significant role in uncertainties of climate models and climate projections (Bony et al., 2015).

One of the extensively discussed topics in analyses of the CMIP6 ensemble is the higher effective climate sensitivity (ECS) of some models and therefore the increased range in ECS now between 1.8 and 5.6 K compared to 2.1 and 4.7 in the CMIP5 ensemble (Meehl et al., 2020; Bock et al., 2020; Schlund et al., 2020). ECS provides a single number, defined as the change in global mean near-surface air temperature resulting from a doubling of the atmospheric $CO_2$ concentration compared to preindustrial conditions, once the climate has reached a new equilibrium (Gregory et al., 2004). A possible reason for the increase of ECS in some models is improvements in cloud representation in these models. Zelinka et al. (2020) show that the increased range of ECS in the CMIP6 models could be explained by an increased range in cloud feedbacks. Studies using single models concluded that the increased climate sensitivity found in these models is largely determined by cloud microphysical processes (Zhu et al., 2022; Frey and Kay, 2018; Gettelman et al., 2019; Bodas-Salcedo et al., 2019). They also point out that the simulated present-day mean state of cloud properties is correlated with the simulated cloud feedback but could also be connected to other coupled feedbacks (Andrews et al., 2019).

As future changes in cloud properties are closely connected to cloud feedbacks and cloud feedbacks are known to be strongly correlated with ECS (see Sect. 3.1), we use ECS as a simple proxy to group the ensemble of CMIP5 and CMIP6 models for this analysis. This facilitates the analysis and allows for obtaining more general conclusions beyond individual models that can vary widely in their sensitivity to the prescribed forcings. A particular focus of this study is whether there are systematic differences in cloud-related quantities between the different ECS groups. The sensitivity of the physical properties to warming is analysed, as this gives some insight into the uncertainty of the projected cloud properties and their potential contribution to cloud feedbacks and ECS.

A qualitative assessment of the present-day model performance by comparing key cloud properties with satellite data is done to help interpreting simulated future changes in cloud properties. We would like to note that conclusions on the plausibility of certain ECS values cannot be drawn from this comparison and are thus not an aim of this study. The performance of CMIP models has also been investigated in other studies. For example, Kuma et al. (2023) applied an artificial neural network to derive cloud types from radiation fields. They found that results from models with a high ECS agree on average better with observations than from models with a low ECS. Jiang et al. (2021) found that the models' ECS is positively correlated with the integrated cloud water content and water vapor performance scores for both CMIP6 and CMIP5 models. In contrast, Brunner et al. (2020) showed that some CMIP6 models with high future warming compared to other models receive systematically

lower performance weights when using anomaly, variance, and trend of surface air temperature, and anomaly and variance of sea level pressure to assess the models' performance.

Here, we first document the differences in the skill of the models in reproducing observed cloud properties among three groups of models sorted by their ECS values and then how the projected changes in cloud properties and cloud radiative effects differ. In Section 2 we briefly introduce the models and observations used as well as the software tool applied to evaluate the models. The representation of cloud properties and cloud radiative effects for all three groups is compared with observational data in Section 3 followed by an analysis of the projected future changes in cloud properties and radiative effects. Section 4 summarizes the discussion and conclusions.

## 2  Data

### 2.1  Models

In this study we use model simulations from the CMIP Phases 5 (Taylor et al., 2012) and 6 (Eyring et al., 2016). The individual models are detailed in Table 1. All model data are freely available via the Earth System Grid Federation (ESGF), which is an international collaboration that manages the decentralized database of CMIP output.

For the analysis presented here, we use historical simulations over the time period 1985–2004 (Table 1) and the scenario simulations of the Representative Concentration Pathway (RCP) 8.5 from CMIP5 and the Shared Socioeconomic Pathways (SSP) 5-8.5 simulations from CMIP6 for the years 2081-2100. The historical simulations use prescribed natural and anthropogenic climate forcings such as concentrations of greenhouse gases and aerosols. We only consider one ensemble member per model, typically the first member "r1i1p1" (CMIP5) and "r1i1p1f1" (CMIP6). As the intermodel spread is typically much larger than the interensemble spread we do not expect our results to change significantly when using more ensemble members for each model. For further details on the model simulations, we refer to Taylor et al. (2012) and Eyring et al. (2016).

ECS and cloud feedbacks are calculated using the simulations forced by an abrupt quadrupling of $CO_2$ (abrupt-4×CO2) and the preindustrial control simulations (piControl) following the method described in Andrews et al. (2012) and Schlund et al. (2020).

In total, the CMIP ensemble investigated here consists of 24 CMIP5 and 27 CMIP6 models that provide the output needed for this analysis. We grouped them into the three groups "low", "medium" and "high" by their ECS values (see Table 1). The thresholds for the three groups are chosen in a way that each of the three groups contains the same number of models. Multi-model group means are calculated as 20-year means over all models in the high, medium and low ECS group applying equal weights to each model.

### 2.2  Observations

The observations and reanalysis data used for the model evaluation are summarized in Table 2. We define one main reference dataset for each variable and for some diagnostics also an alternative reference dataset. The time period covered depends on the

**Table 1.** List of CMIP5 and CMIP6 models grouped by ECS value into three roughly equally sized groups "high" (ECS > 4.0 K), "medium" (2.87 K < ECS < 4.0 K) and "low" (ECS < 2.87 K).

| Number | CMIP5 model | CMIP6 model | ECS (K) | Citation |
|---|---|---|---|---|
| 1 | | CanESM5 | 5.62 | Swart et al. (2019) |
| 2 | | HadGEM3-GC31-LL | 5.55 | Williams et al. (2018); Kuhlbrodt et al. (2018) |
| 3 | | HadGEM3-GC31-MM | 5.42 | Williams et al. (2018); Kuhlbrodt et al. (2018) |
| 4 | | UKESM1-0-LL | 5.34 | Sellar et al. (2019) |
| 5 | | CESM2 | 5.16 | Danabasoglu et al. (2020) |
| 6 | | CNRM-CM6-1 | 4.83 | Voldoire et al. (2019) |
| 7 | | KACE-1-0-G | 4.77 | Lee et al. (2020a) |
| 8 | | CNRM-ESM2-1 | 4.76 | Séférian et al. (2019) |
| 9 | | CESM2-WACCM | 4.75 | Danabasoglu et al. (2020) |
| 10 | | NESM3 | 4.72 | Cao et al. (2018) |
| 11 | MIROC-ESM | | 4.67 | Watanabe et al. (2011) |
| 12 | HadGEM2-ES | | 4.61 | Collins et al. (2011) |
| 13 | | IPSL-CM6A-LR | 4.56 | Boucher et al. (2020) |
| 14 | | TaiESM1 | 4.31 | Lee et al. (2020b) |
| 15 | IPSL-CM5A-LR | | 4.13 | Dufresne et al. (2013) |
| 16 | IPSL-CM5A-MR | | 4.12 | Dufresne et al. (2013) |
| 17 | CSIRO-Mk3-6-0 | | 4.08 | Rotstayn et al. (2010) |
| 1 | GFDL-CM3 | | 3.97 | Donner et al. (2011) |
| 2 | BNU-ESM | | 3.92 | Ji et al. (2014) |
| 3 | ACCESS1-0 | | 3.83 | Bi et al. (2013) |
| 4 | CanESM2 | | 3.69 | Arora et al. (2011) |
| 5 | MPI-ESM-LR | | 3.63 | Giorgetta et al. (2013); Stevens et al. (2013) |
| 6 | | CMCC-ESM2 | 3.58 | Cherchi et al. (2019) |
| 7 | ACCESS1-3 | | 3.53 | Bi et al. (2013) |
| 8 | | CMCC-CM2-SR5 | 3.52 | Cherchi et al. (2019) |
| 9 | MPI-ESM-MR | | 3.46 | Giorgetta et al. (2013); Stevens et al. (2013) |
| 10 | FGOALS-g2 | | 3.38 | Li et al. (2013) |
| 11 | | MRI-ESM2-0 | 3.15 | Yukimoto et al. (2019); Mizuta et al. (2012) |
| 12 | | GISS-E2-1-H | 3.11 | Kelley et al. (2020) |
| 13 | | BCC-CSM2-MR | 3.04 | Wu et al. (2019) |
| 14 | | FGOALS-f3-L | 3.00 | He et al. (2020) |
| 15 | | MPI-ESM1-2-LR | 3.00 | Mauritsen et al. (2019) |
| 16 | | MPI-ESM1-2-HR | 2.98 | Muller et al. (2018) |
| 17 | CCSM4 | | 2.94 | Gent et al. (2011) |
| 18 | | FGOALS-g3 | 2.88 | Li et al. (2020b) |
| 1 | bcc-csm1-1-m | | 2.86 | Wu et al. (2010); Wu (2012) |
| 2 | bcc-csm1-1 | | 2.83 | Wu et al. (2010); Wu (2012) |
| 3 | NorESM1-M | | 2.80 | Bentsen et al. (2013) |
| 4 | | GISS-E2-1-G | 2.72 | Kelley et al. (2020) |
| 5 | MIROC5 | | 2.72 | Watanabe et al. (2010) |
| 6 | | MIROC-ES2L | 2.68 | Hajima et al. (2020) |
| 7 | | MIROC6 | 2.61 | Tatebe et al. (2019) |
| 8 | IPSL-CM5B-LR | | 2.60 | Hourdin et al. (2013) |
| 9 | MRI-CGCM3 | | 2.60 | Yukimoto et al. (2012) |
| 10 | | NorESM2-LM | 2.54 | Seland et al. (2020) |
| 11 | | NorESM2-MM | 2.50 | Seland et al. (2020) |
| 12 | GFDL-ESM2M | | 2.44 | Donner et al. (2011) |
| 13 | GFDL-ESM2G | | 2.39 | Donner et al. (2011) |
| 14 | GISS-E2-H | | 2.31 | Schmidt et al. (2006) |
| 15 | | CAMS-CSM1-0 | 3.29 | Rong et al. (2018) |
| 16 | GISS-E2-R | | 2.11 | Schmidt et al. (2006) |
| 17 | inmcm4 | | 2.08 | Volodin et al. (2010) |

**Table 2.** List of observational and reanalysis datasets and time periods used for the model evaluation.

| Variable | Reference Dataset | Alternative Reference Dataset |
|---|---|---|
| Total Cloud Fraction | ESACCI Cloud, 1992-2016 (Stengel et al., 2020) | MODIS, 2003-2018 (Platnick et al., 2003) |
| Ice Water Path, Liquid Water Path | ESACCI Cloud, 1992-2016 (Stengel et al., 2020) | CloudSat, 2006-2017 (Stephens et al., 2018) |
| Cloud Radiative Effect, TOA Outgoing Radiation | CERES-EBAF Ed4.2, 2001-2022 (Loeb et al., 2018; Kato et al., 2018) | ESACCI Cloud, 1992-2016 (Stengel et al., 2020) |
| Temperature | ERA5, 1985-2004 (, C3S) | NCEP, 1985-2004 (Kalnay et al., 1996) |
| Precipitation | GPCP-SG, 1985-2004 (Adler et al., 2003; Huffman and Bolvin, 2013) | ERA5, 1985-2004 (, C3S) |

data availability for the specific reference dataset and is given in 2. We would like to note that the time period from the models used for comparison with the observations (see Sect. 2.1) does not match exactly the observed years. It is not surprising, however, that this has very little impact on the multi-year group averages as ESMs are not expected to reproduce the exact 95 observed phase of climate modes largely controlling present-day variability of clouds but rather their statistical properties.

As reference dataset for cloud parameters like cloud cover and ice and liquid water path we use the cloud product version 3 of the European Space Agency Climate Change Initiative (ESACCI-CLOUD) which is based on data from the passive imager satellite sensors Advanced Very High Resolution Radiometer (AVHRR) and (A)ATSR. For more details we refer to Stengel et al. (2020). The alternative reference dataset for cloud cover is the Moderate Resolution Imaging Spectroradiometer (MODIS) 100 L3 Atmosphere Product (Platnick et al., 2003) and for ice and liquid water path we used cloud profiling radar measurements from CloudSat (Stephens et al., 2018). The Clouds and Earth's Radiant Energy Systems (CERES) Energy Balanced and Filled (EBAF) Ed4.1 dataset (Loeb et al., 2018; Kato et al., 2018) provides global monthly mean top of atmosphere (TOA) longwave (LW), shortwave (SW), and net radiative fluxes under clear-sky and all-sky conditions, which are used as a reference dataset to calculate the cloud radiative effects and the TOA outgoing radiation. CERES instruments are flown on NASA's Terra and 105 Aqua satellites.

For temperature, we use data from the European Centre's for Medium-Range Weather Forecasts (ECMWF) fifth-generation reanalysis ERA5 (, C3S) and the National Centers' for Environmental Prediction (NCEP) Reanalysis Project (Kalnay et al., 1996). Data from the Global Precipitation Climatology Project Satellite-Gauge Combination (GPCP-SG) Version 2.3 (Adler et al., 2003) and ERA5 are used as reference datasets for precipitation.

**2.3 ESMValTool**

All analyses in this study are performed with the open-source community diagnostics and performance metrics tool for evaluation of ESMs "Earth System Model Evaluation Tool" (ESMValTool) version 2 (Eyring et al., 2020; Lauer et al., 2020; Righi et al., 2020; Weigel et al., 2021). All figures from this paper can be reproduced by running the ESMValTool "recipe" (con-

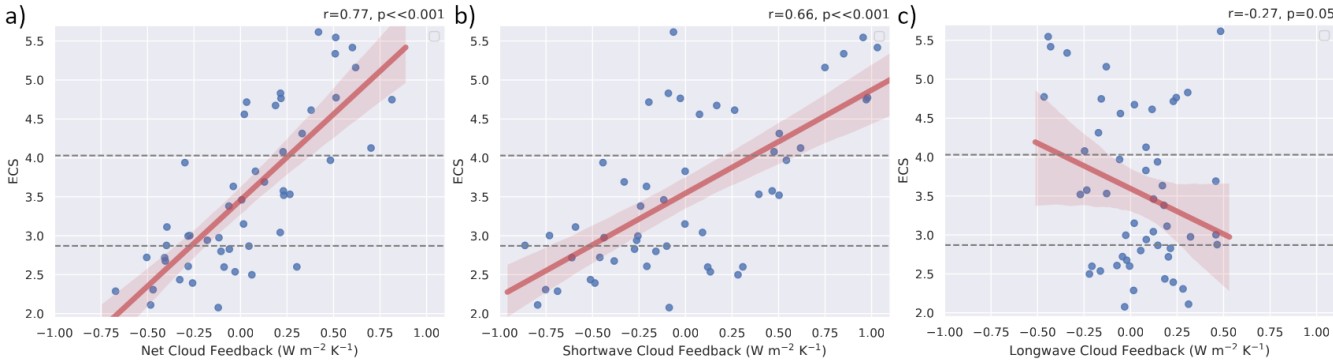

**Figure 1.** Scatterplot of the global mean a) net, b) shortwave and c) longwave cloud feedback (x-axis) and ECS (y-axis) of the CMIP models (Table 1) with regression line including the confidence interval of the regression of 95%. Horizontal dashed lines indicate separations of the three ECS groups (see Table 1).

figuration script defining all datasets, processing steps and diagnostics) recipe_bock23acp.yml (see also the Code and Data

Availability Section).

## 3    Analysis

### 3.1    ECS and cloud feedback

The large spread in ECS of CMIP6 models could be mainly explained by uncertainties in the simulated net cloud feedback. The net cloud feedback is defined as change in the sum of shortwave and longwave cloud radiative effects at the top of the

atmosphere (TOA) per degree of surface warming (2-m temperature) calculated from abrupt-4×CO2 simulations compared to the corresponding piControl simulations. The net cloud feedback is typically dominated by the shortwave component (Zelinka et al., 2020).

The relation between ECS and simulated cloud feedbacks is illustrated in Figure 1, which shows the correlation between net, shortwave and longwave cloud feedbacks and ECS in the CMIP5 and CMIP6 models (Table 1). The relation between net

cloud feedback and ECS is dominated by the shortwave cloud feedback, which shows a strong correlation with ECS (r = 0.66 and a small p value of p = 3.6e-9). For the longwave cloud feedback there is only a weak (negative) correlation with ECS (p = 0.05).

As the representation of clouds and their sensitivity to climate change have a strong impact on the ECS (Zelinka et al., 2020; Bjordal et al., 2020; Bony et al., 2015) and because the range of ECS obtained from the ensemble of CMIP6 models is

larger than the one from the previous model generations (Meehl et al., 2020), this motivated us to look into the differences in present-day performance and future projections of physical cloud parameters from models with low/medium/high ECS.

Figure 2 shows the geographical distributions of the net, shortwave and longwave cloud feedbacks averaged over all models within each group. The pattern of the net cloud feedback is dominated by the geographical distribution of the shortwave cloud

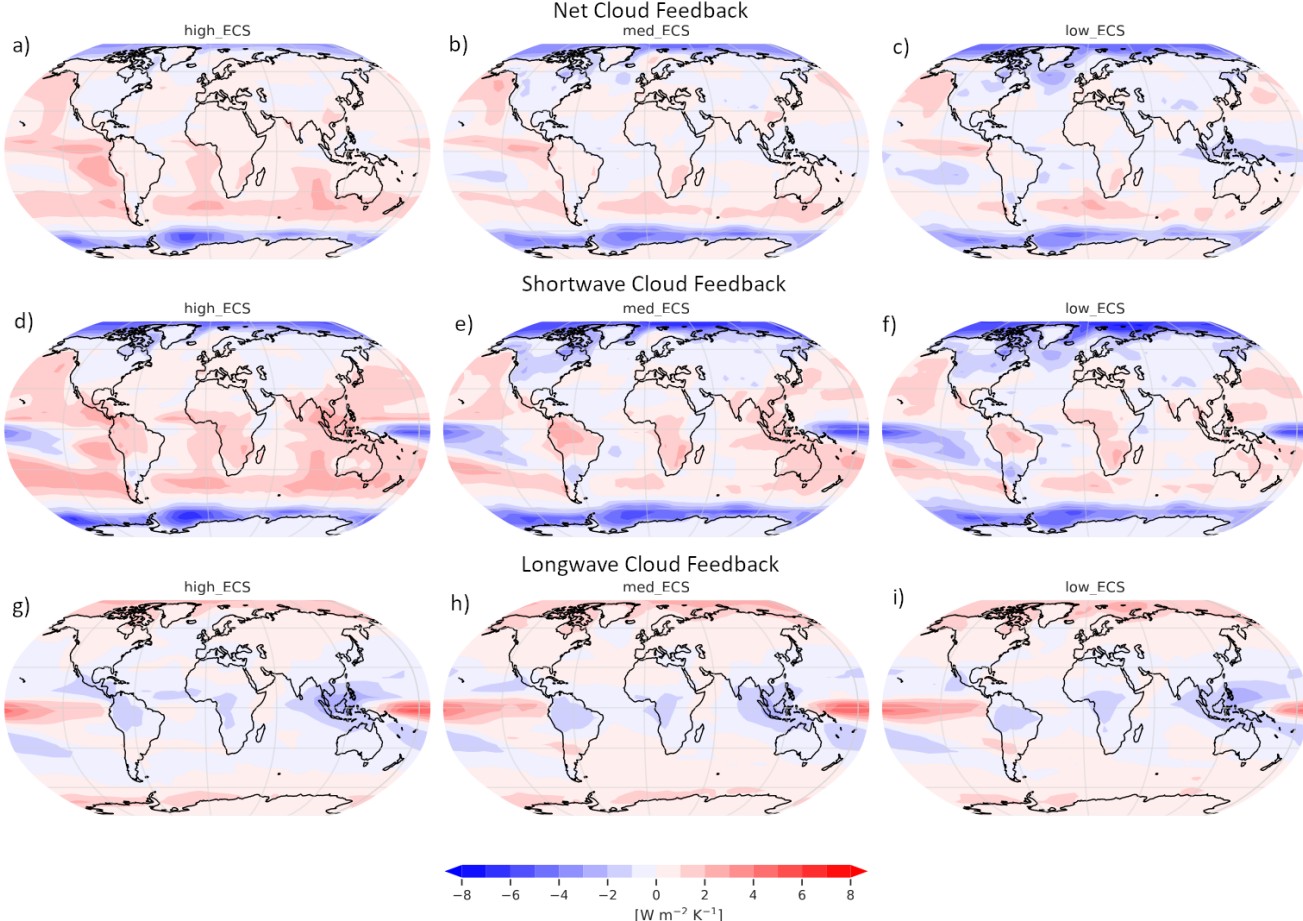

**Figure 2.** Geographical maps of net (a,b,c), shortwave (d,e,f) and longwave (g,h,i) cloud feedback for high (left), medium (middle) and low (right) ECS groups.

feedback. On global average, the high ECS group has the largest net cloud feedback of 0.41 W m$^{-2}$, followed by the medium ECS group (0.01 W m$^{-2}$) and the low ECS group (-0.20 W m$^{-2}$). The group mean net cloud feedback changes sign at around 60°S and 80°N in all three groups. The sign change at around 60°S in the shortwave cloud feedback indicates where the models are switching from clouds with an ice component in the piControl simulations to clouds consisting almost entirely of liquid droplets in the abrupt-4xCO2 experiment (Ceppi et al., 2017). With increasing latitude there is an increasing ice fraction in the model clouds that supports a negative shortwave feedback as cloud particles can change phase with warming. Particularly over the Arctic and the tropical Pacific, the (negative) shortwave cloud feedback is partly or fully compensated by a (positive) longwave cloud feedback resulting in rather small net cloud feedback values.

The high ECS models show a more positive net cloud feedback in the Tropics and midlatitudes, especially over the Southern Ocean, than the other two groups. The group mean of the low ECS models shows a distinct negative net cloud feedback in

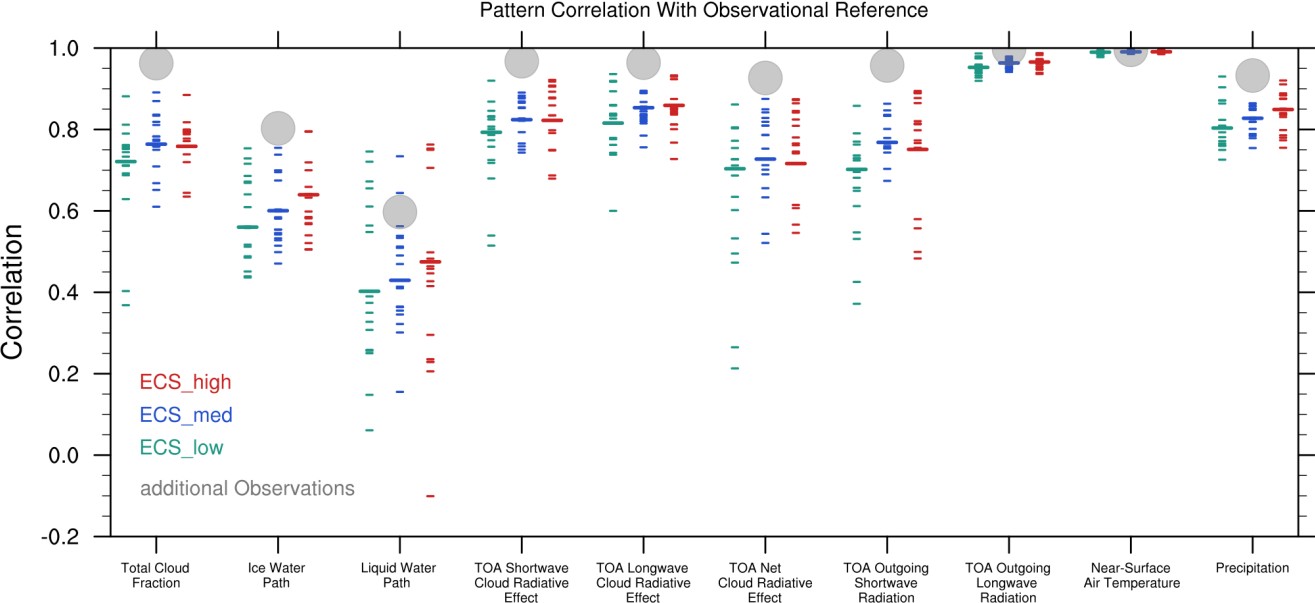

**Figure 3.** Centered pattern correlations between models and observations for the annual mean climatology over the time period 1985–2004. Results are shown for the individual CMIP models as short lines, along with the corresponding group averages (long lines) for the three ECS groups. The correlations are shown between the models and the reference datasets listed in Table 2. In addition, the correlations between the reference and alternative reference datasets are shown (solid gray circles). To ensure a fair comparison across a range of model resolutions, the pattern correlations are computed after regridding all datasets to a common resolution of 2° in longitude and 2° in latitude and applying a common missing value masking.

the Tropics, particularly in the tropical Pacific. This signal is much weaker in the other two groups. The reason is a more

pronounced negative shortwave cloud feedback particularly over the Pacific Intertropical Convergence Zone (ITCZ) and South Pacific Convergence Zone (SPCZ) in the group mean of the low ECS models.

## 3.2  Evaluation of cloud properties

The modeled mean state of cloud properties in ESMs is correlated with the simulated simulated cloud feedback (Zelinka et al., 2020). We therefore evaluate the cloud climatologies from the three ECS groups by comparing the model results with satellite

observations and reanalysis datasets. In order to get an overview on the performance of the three model groups in reproducing observed cloud properties, we calculate the centered pattern correlations of selected cloud properties and cloud radiative effects with satellite observations and reanalyses (Figure 3) for all individual models as well as for the group means.

For most of the variables investigated such as ice water path, cloud radiative effects and precipitation, the high ECS models show a better agreement (i.e. higher pattern correlation) with observations than the two other groups. Little differences are

found for fields that are quite well simulated by all models such as near-surface air temperature and TOA outgoing longwave

radiation. It is noteworthy that except for total cloud cover, the worst-performing models for all variables investigated are found within the low ECS groups.

The observed geographical patterns of the multi-year annual mean total cloud cover, ice water path and liquid water path are relatively poorly reproduced by all three model groups with large intermodel spreads (Figure 3). For cloud ice and cloud liquid water path the pattern correlations between ESACCI Cloud (passive instrument) and the alternative measurements of CloudSat (active instrument) show the large uncertainties of these quantities derived from satellite observations (e.g., Lauer et al., 2023). An additional uncertainty in this comparison is introduced, as some CMIP models may provide the sum of cloud ice and falling ice (e.g. snow) in the ice water path values if the falling ice is included in their radiation calculations. The number of models including falling ice radiative effects, however, is rather small and thus not expected to play an important role in the group means. An overview can be found e.g. in Li et al. (2020a), their Table 1. The ice water path from the high ECS models has a noticeably smaller intermodel spread (1 sigma = 0.07 kg m$^{-2}$ compared with 0.12 kg m$^{-2}$ from the low ECS group). Similarly, the range of results from the low ECS group for the cloud radiative effects is larger than for the two other groups.

In order to investigate possible reasons for these differences among the three ECS groups, we compare the geographical distributions of the cloud properties for each individual group to climatologies from satellite observations (Figures 4, 5, 6 and 7). Here, we focus on the most climate relevant parameters, which are available from both models and satellite observations. These are total cloud fraction, liquid water and ice water path and cloud radiative effects (longwave, shortwave, net). For the comparison, output from satellite simulators such as the Cloud Feedback Model Intercomparison Project (CFMIP) Observation Simulator Package COSP (Bodas-Salcedo et al., 2011) can make the model results more directly comparable to the satellite data. Applied online during the model simulations, COSP is able to mimic the satellite viewing geometry, temporal sampling, and specific instrument characteristics such as cutoff values for some cloud related quantities. Most of the CMIP5 and CMIP6 historical simulations, however, do not provide such output. Of all variables investigated here, only total cloud fraction is available, other variables from satellite simulators such as cloud liquid and ice water path or radiative fluxes are not available for these models. Restricting our analysis on the available output from satellite simulators would therefore reduce the sample size of the three different ECS groups to a degree, where any differences among the groups are expected to be purely random. In the following, we therefore use the 'native' model output for comparison. We would like to note that this limitation restricts a quantitative assessment of differences between models and observations as an unknown error is introduced by comparing two not fully equal quantities regarding their definition (e.g. observational thresholds) as well as temporal and spatial sampling. An assessment of the present-day model performance beyond a qualitative analysis to help interpreting simulated future changes in cloud properties or an assessment of differences in the plausibility of certain ECS values is therefore not possible.

**Total cloud fraction**

The annual mean total cloud fraction from ESACCI Cloud (Figure 4a) shows the known geographical patterns: maxima over land in the Tropics due to strong convection, minima in the subtropics because of descending air with local maxima in stratocumulus regions off the west coasts of the continents (Africa, North and South America), maxima in the midlatitudes over the ocean especially over the Southern Ocean and minima over polar regions where the air is very cold and dry.

## Total Cloud Fraction

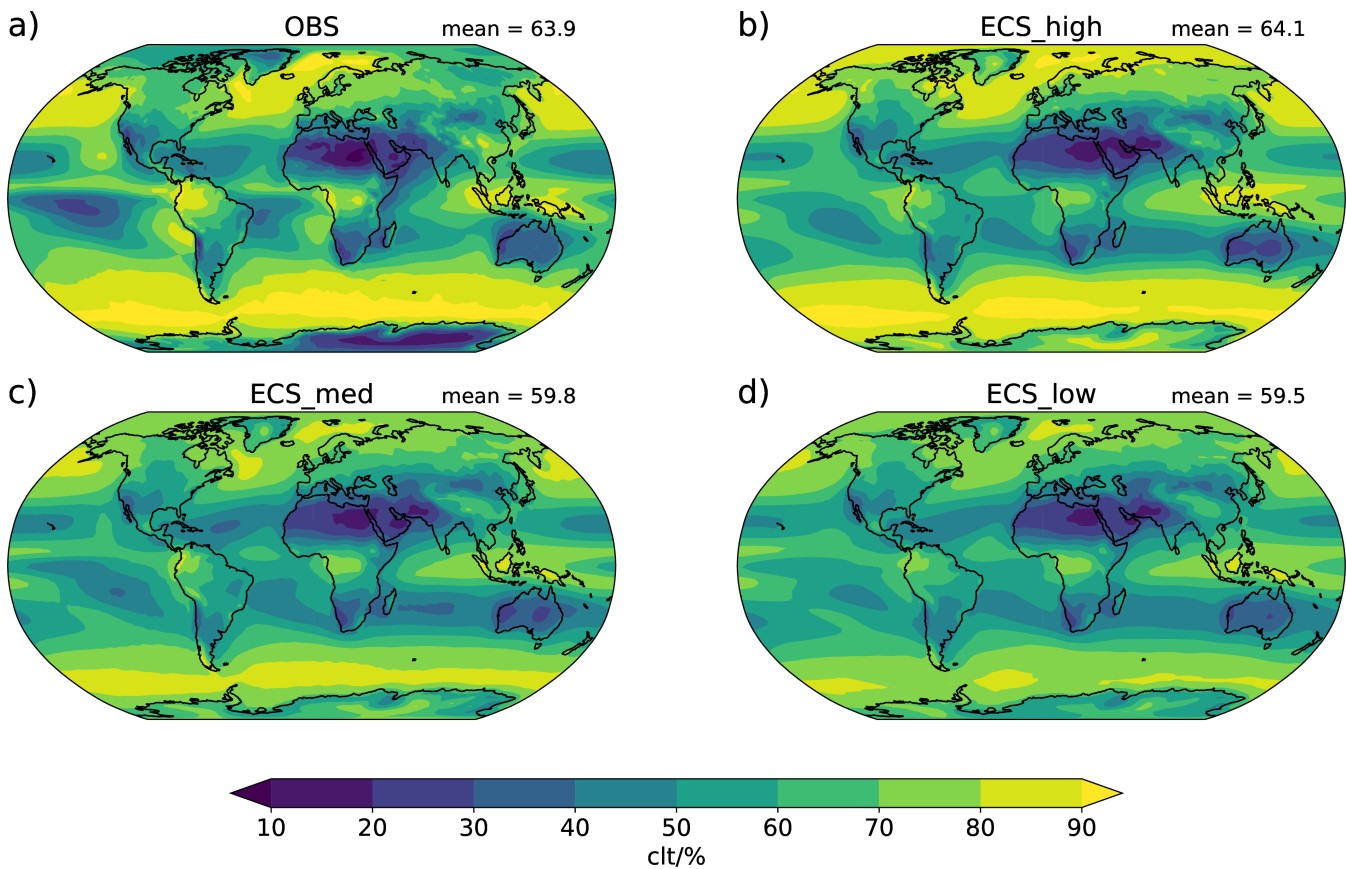

**Figure 4.** Geographical map of the multi-year annual mean total cloud fraction from (a) ESACCI Cloud (OBS) and (b,c,d) group means of historical CMIP simulations from all three ECS groups.

The group mean of the high ECS models (Figure 4a) shows a smaller global mean bias of 0.2% in total cloud cover compared to about -4% from the two other groups as well as a smaller root mean square difference (RMSD) of 10.0% with an estimated uncertainty range of 9.9 to 10.3% than the other two groups (group average RMSD = 10.4 for the medium models and 11.5% for the low models; see Table 3). While the group mean of the high ECS models lies in between the observational range of global mean total cloud cover, the root mean square deviation (RMSD) from all models and group means exceeds significantly the observational uncertainty of the ESACCI Cloud dataset which is estimated to be 3% (Lauer et al., 2023). The pattern correlations of the high and medium ECS group mean are slightly higher than of the low ECS group (see Table 4). The interquartile of all high ECS model correlations in respect to the reference (ESACCI Cloud) also lies above the ones of the low ECS models. But all correlation values of the models are clearly smaller than the observational uncertainty where correlation values are ranging from 0.96 to 0.99 (Lauer et al., 2023). In the midlatitudes the high ECS models with a typical bias of less

**Table 3.** Mean values and root mean square difference of each group mean together with the 25% and 75% quantiles in parenthesis calculated by bootstrapping (1000 times, sample size = number of models in the group). The second line gives the 25% and 75% quantiles calculated from all individual models. The RMSD values are calculated in comparison to the corresponding reference dataset (see Table 2, second column).

| Variable | Mean | | | RMSD | | |
|---|---|---|---|---|---|---|
| | high ECS | med ECS | low ECS | high ECS | med ECS | low ECS |
| Total Cloud Fraction (%) | 64.1 (63.3, 65.0) | 59.8 (58.9, 60.0) | 59.5 (59.0, 59.8) | 10.0 (9.9, 10.3) | 10.4 (10.1, 10.9) | 11.5 (11.2, 12.0) |
| | (61.9, 68.8) | (56.7, 62.5) | (57.8, 61.9) | (12.8, 13.8) | (11.1, 15.0) | (12.3, 15.6) |
| Ice Water Path (g m$^{-2}$) | 37.0 (34.3, 40.1) | 34.6 (30.3, 38.6) | 40.7 (35.5, 45.2) | 36.0 (35.0, 37.2) | 34.2 (31.1, 38.8) | 30.5 (29.1, 34.5) |
| | (19.1, 51.9) | (17.6, 40.6) | (14.9, 42.3) | (37.5, 51.0) | (41.4, 56.3) | (38.0, 56.0) |
| Liquid Water Path (g m$^{-2}$) | 65.0 (61.0, 68.5) | 72.1 (67.1, 76.8) | 83.2 (78.5, 87.9) | 37.1 (33.9, 40.6) | 41.5 (38.1, 45.6) | 49.1 (45.2, 53.7) |
| | (55.3, 68.4) | (54.6, 86.1) | (60.4, 105.5) | (34.3, 46.7) | (35.4, 57.0) | (38.5, 78.6) |
| Net Cloud Radiative Effect (W m$^{-2}$) | -22.8 (-23.4, -22.3) | -23.2 (-23.6, -22.8) | -25.8 (-26.3, -25.3) | 9.3 (9.0, 9.9) | 9.2 (9.0, 9.7) | 12.3 (11.7, 13.0) |
| | (-24.7, -20.7) | (-25.0, -21.9) | (-28.2, -23.6) | (9.5, 14.1) | (10.5, 13.0) | (11.9, 18.3) |

**Table 4.** Pattern correlation of each group mean together with the 25% and 75% quantiles in parentheses calculated by bootstrapping (1000 times, sample size = number of models in the group). The second line gives the 25% and 75% quantiles calculated from all individual models. The correlation is calculated in comparison to the corresponding reference dataset (see Table 2, second column).

| Variable | Correlation | | |
|---|---|---|---|
| | high ECS | med ECS | low ECS |
| Total Cloud Fraction | 0.84 (0.83, 0.84) | 0.85 (0.84, 0.85) | 0.81 (0.79, 0.82) |
| | (0.77, 0.80) | (0.75, 0.83) | (0.70, 0.76) |
| Ice Water Path | 0.63 (0.61, 0.64) | 0.83 (0.77, 0.82) | 0.76 (0.71, 0.78) |
| | (0.56, 0.65) | (0.53, 0.68) | (0.49, 0.68) |
| Liquid Water Path | 0.50 (0.45, 0.56) | 0.55 (0.52, 0.57) | 0.58 (0.54, 0.60) |
| | (0.25, 0.50) | (0.36, 0.54) | (0.27, 0.60) |
| Net Cloud Radiative Effect | 0.86 (0.84, 0.86) | 0.86 (0.85, 0.87) | 0.79 (0.75, 0.80) |
| | (0.74, 0.84) | (0.55, 0.77) | (0.55, 0.77) |

than -5% are in better agreement with the observations than the group means from the two other ECS groups (-10 to -15%). Especially the maxima in total cloud cover over the Southern Ocean and the northern Atlantic (Figure 4b) are better represented in the group mean of the high ECS group (0 to -5%) where the known bias of CMIP models is reduced (Lauer et al., 2023). In contrast, the minima over the polar regions seen in ESACCI Cloud are better reproduced by the low and medium ECS models (Figure 4d) (bias 20 to 40%) than in the high ECS group (bias 30 to 60%). We would like to note, however, that many satellite products based on passive instruments such as ESACCI have difficulties in detecting optically thin clouds (e.g., Karlsson et al., 2017). Total cloud cover from these instruments can therefore be assumed to be significantly biased low in the polar regions.

# Ice Water Path

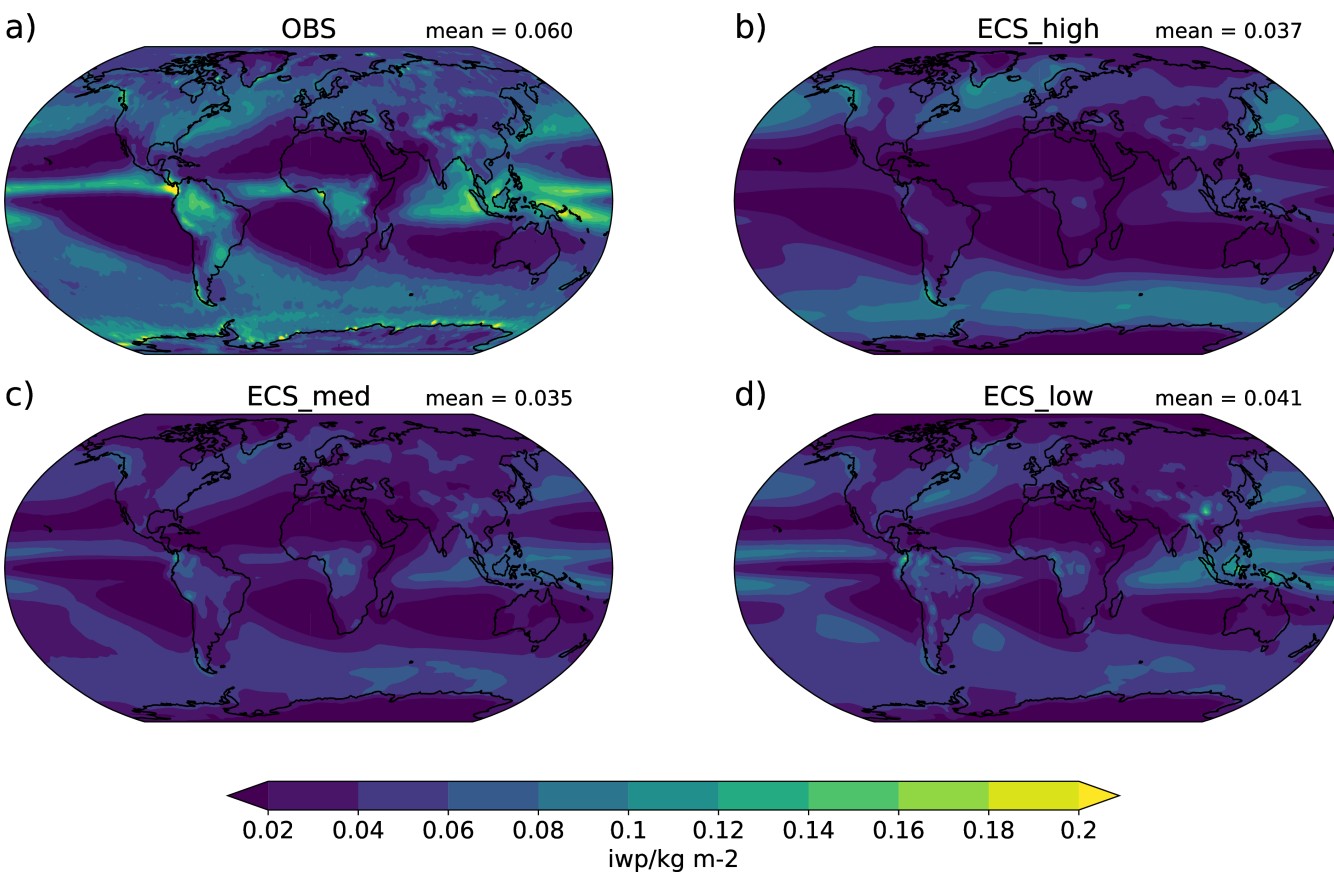

**Figure 5.** Same as Figure 4 but for cloud ice water path.

## Ice water path

The global distribution of ice water path (Figure 5a) from ESACCI Cloud shows a maximum in the ITCZ due to frequent convection of up to 0.2 kg m$^{-2}$. The absolute minima of ice water path are found in the subtropics in the subsidence regions west of continents. High amounts of cloud ice are also found along the stormtracks in midlatitudes, with values decreasing towards the poles.

All three groups of CMIP models underestimate the observed amount of ice water path with global mean biases of almost 40%. The global mean bias is quite similar among the different ECS groups and there are no significant differences between the mean values from the three model groups (Table 3). We would like to note that the global average ice water path from the ESACCI Cloud dataset used as a reference is at the upper range of satellite observations (36 to 61 g m$^{-2}$, Lauer et al. (2023)) whereas the group means are at the lower end of this range. The RMSD of 36.0 g m$^{-2}$ and the correlation of 0.63 of the high ECS group mean are the worst of all groups. In contrast, the correlation values of the medium and low ECS group

means are within the observational range of 0.74 and 0.93 (Lauer et al., 2023). One reason for this is that the observed high ice water path values in the Tropics related to the ITCZ are the least underestimated in the low ECS group. In contrast, the observed maxima in midlatitudes, especially over the Southern Ocean, are best reproduced by the high ECS models (Figure 5b). This is consistent with the group mean performance for total cloud fraction and supports the hypothesis that the improved representation of supercooled liquid in some of the high ECS models (leading to better agreement with observations) leads to a higher ECS as it decreases the magnitude of a negative cloud phase feedback (e.g., Bock et al., 2020; Zelinka et al., 2020; Bjordal et al., 2020; Frey and Kay, 2018).

**Liquid water path**

ESACCI Cloud satellite observations of cloud liquid water path (Figure 6a) show local maxima in the ITCZ and the stratocumulus regions in the subtropics. The largest values of liquid water path are found in the extratropics in the stormtrack regions mainly over the Southern Ocean and the northern Atlantic.

There is a positive bias in liquid water path in all three model groups ranging from a global average of 20.2 g m$^{-2}$ (44%) in the high ECS group to 38.4 g m$^{-2}$ (85%) in the low ECS group. Satellite measurements of the cloud liquid water are known to suffer from a high degree of uncertainty (e.g., Lauer et al., 2023). The group means are therefore all in between the observed range of 36 to 105 g m$^{-2}$ (Lauer et al., 2023) making an assessment of the model performance difficult. Regarding the RMSD the high ECS group mean performs with 37.1 g m$^{-2}$ better than the other two groups (medium group mean with 41.5 g m$^{-2}$ and high group mean with 49.1 g m$^{-2}$) (see Table 3). For RMSD, all three group means exceed the observational uncertainty estimate for ESACCI Cloud (30 g m$^{-2}$; Lauer et al. (2023)). All three groups show poor correlations (see Table 4). The low ECS models are at the lower end of the observational range (0.49 to 0.94). All three group means show a higher cloud liquid water path in the ITCZ and in the midlatitude storm track regions than the observations. The local maxima in the stratocumulus regions seen in the observations are underestimated in all three group means and related to the known bias of underestimating the cloud fraction of stratocumulus clouds in the CMIP models (e.g., Jian et al., 2020).

**Cloud radiative effects**

The cloud radiative effects are calculated as the differences in top of the atmosphere clear-sky and all-sky radiative fluxes. The net cloud radiative effect is the sum of the negative (cooling) shortwave and a positive (warming) longwave cloud radiative effect. The ESACCI Cloud observations show a global mean net cooling due to clouds of about -21 W m$^{-2}$ (Figure 7a). Clouds have a warming radiative effect in particular over regions with a high surface albedo like ice covered regions in Greenland and Antarctica and the desert regions in North Africa. A large negative net radiative effect of clouds is found over the stratocumulus regions in the subtropics and in the midlatitude stormtrack regions. In the ITCZ there is a partly compensating effect between the shortwave and longwave radiative effects leading to smaller absolute net values than in the stratocumulus and stormtrack regions.

The amplitude of the global mean net cloud radiative effect is slightly overestimated in the models with the largest bias in the low ECS group (mean bias = -4.8 W/m$^2$, RMSD = 9.0 W/m$^2$) and the smallest bias in the high ECS group (mean bias = -1.8

## Liquid Water Path

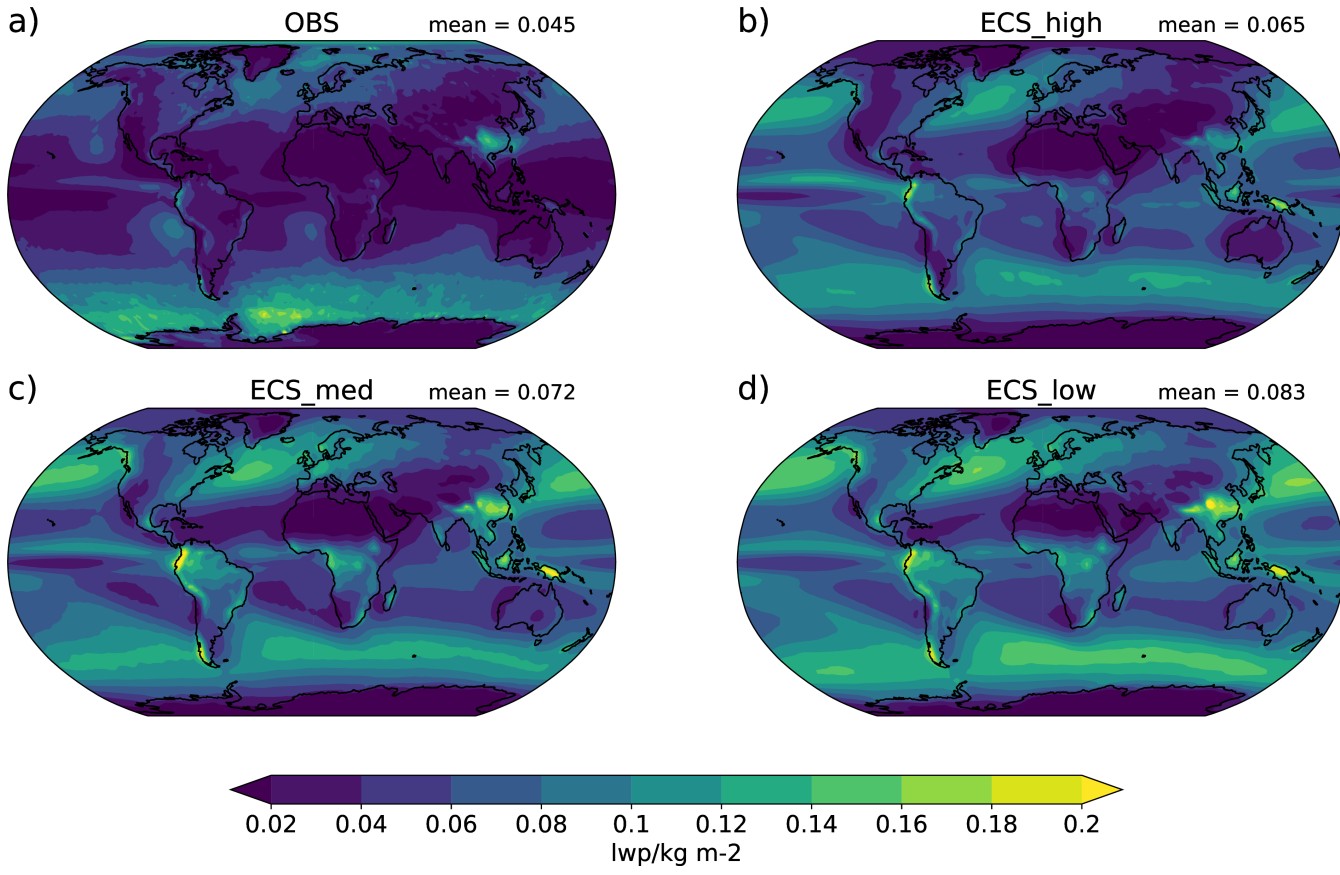

a) OBS     mean = 0.045

b) ECS_high     mean = 0.065

c) ECS_med     mean = 0.072

d) ECS_low     mean = 0.083

0.02   0.04   0.06   0.08   0.1   0.12   0.14   0.16   0.18   0.2

lwp/kg m-2

**Figure 6.** Same as Figure 4 but for cloud liquid water path.

W/m$^2$, RMSD = 6.5 W/m$^2$) (see also Table 3). While the global mean biases of the group means are within the observational uncertainty range, the RMSD values are larger than the ones of different individual observational datasets when compared to a reference dataset consisting of an average over different products (Lauer et al., 2023). The geographical patterns of the three model groups agree well with the ESACCI Cloud observations (Figure 7). The linear pattern correlations of the annual average net cloud radiative effect from the high ECS group mean with observations is slightly higher (0.91) than with the medium (0.90) and low (0.86) ECS group. This is also reflected in the range of correlation values from the individual models in each group given by the 25% and 75% quantiles. These range between 0.66 and 0.79 in the low ECS group, between 0.73 and 0.85 in the medium ECS group and between 0.79 and 0.89 in the high ECS group. For comparison, the range of correlation coefficients of different observational datasets is 0.98-0.99 (Lauer et al., 2023). The peaks of positive cloud forcing over land over Greenland, North Africa and the west coast of North and South America are underestimated in all three groups. In these regions, however, observational uncertainties are expected to be large because of high surface albedo, topography or very low

## TOA Net Cloud Radiative Effect

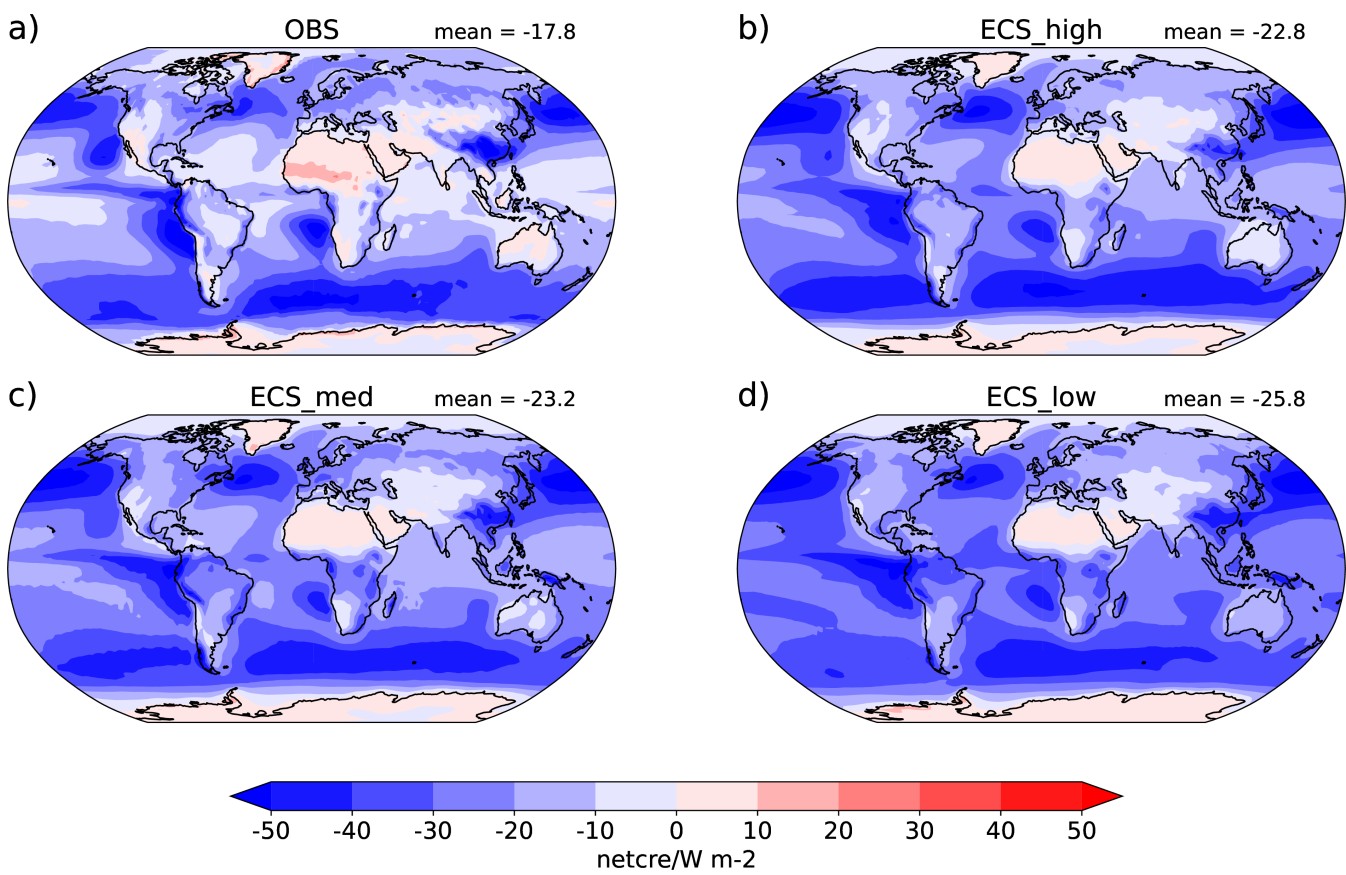

**Figure 7.** Same as Figure 4 but for net cloud radiative effect.

cloud cover. The largest positive bias for all groups is found over the stratocumulus regions with up to 46 W/m$^2$ locally. Apart from this, the low ECS group shows particularly between 30°S and 30°N (Figure 7d), a too strong net cloud radiative effect resulting mainly from a too strong shortwave cooling of the clouds in this latitude belt (Figure 9e) seemingly caused by the largest cloud water path values of all three ECS groups (Figure 9b,c).

### 3.3 Differences in projected future cloud properties

In order to investigate the sensitivity of cloud parameters simulated by the three ECS groups to future warming, we compare the changes in selected cloud properties and cloud radiative effects in future simulations from each group. For CMIP6 we calculate the changes as differences between data from SSP5-8.5 and for CMIP5 from RCP8.5 and results to the respective historical simulations.

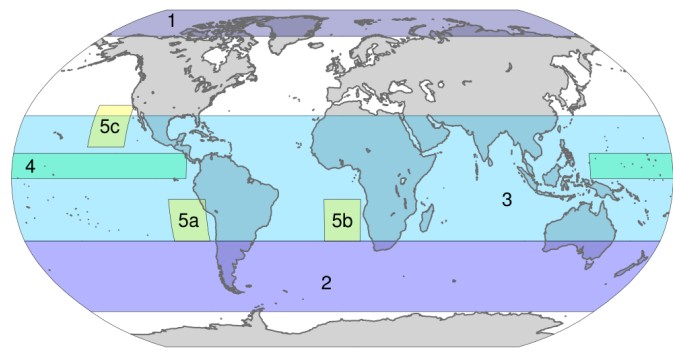

**Figure 8.** Maps of selected regions: 1) Arctic (70-90°N), 2) Southern Ocean (30-65°S), 3) Tropical Ocean (30°N-30°S) and 4) Pacific ITCZ (0-12°N, 135°O-85°W) and Figure 11: 5a) South East Pacific (10-30°S, 75-95°W), 5b) South East Atlantic (10-30°S, 10°W-10°O) and 5c) North East Pacific (15-35°N, 120-140°W)

The zonally averaged group means (Figure 9a-f, upper panels) show the results from the historical and the scenario simulations for the investigated cloud properties (total cloud fraction, ice and liquid water path and cloud radiative effects) for the different ECS groups. Projected zonal mean changes per degree warming (near-surface temperature increase) are displayed in the panels below (Figure 9a-f, lower panels). Additionally, we show the sensitivity of cloud parameters from each ECS group

over the ocean for selected regions. The relative changes (calculated as the differences between the scenario value and the historical value divided by the historical value) in cloud parameters per degree warming averaged over selected regions (Figure 8) are shown in Figure 10: 1) Arctic, 2) Southern Ocean, 3) tropical ocean and 4) Pacific ITCZ and Figure 11: 5a) South East Pacific, 5b) South East Atlantic and 5c) North East Pacific.

In the following, we discuss in more detail differences in projected future cloud properties for each cloud parameter.

**Total cloud cover**

For zonal mean cloud cover (Figure 9a), the comparison of the historical runs with the scenario simulations shows an increase in the zonal mean cloud cover in particular over the polar regions north and south of about 70°. This positive sensitivity to warming shows maximum values ranging between about 0.5%/K for the high, about 1%/K for the medium and 1.4%/K for the low ECS groups.

Particularly in the Tropics and in SH mid- and high latitudes, the sensitivity of simulated cloud cover to warming is quite different among the high ECS group and the two other groups. While the low and medium ECS groups show a mostly positive sensitivity in the Tropics, the high ECS group shows a negative sensitivity of cloud cover to warming of about 0.5 to -1.5%/K. Averaged over the tropical ocean (Figure 10c), the behavior of the high ECS models is significantly different than that of the two other groups. All high ECS models show a decrease in total cloud cover over the tropical ocean while the individual models

in the two other groups do not agree on the sign of the change.

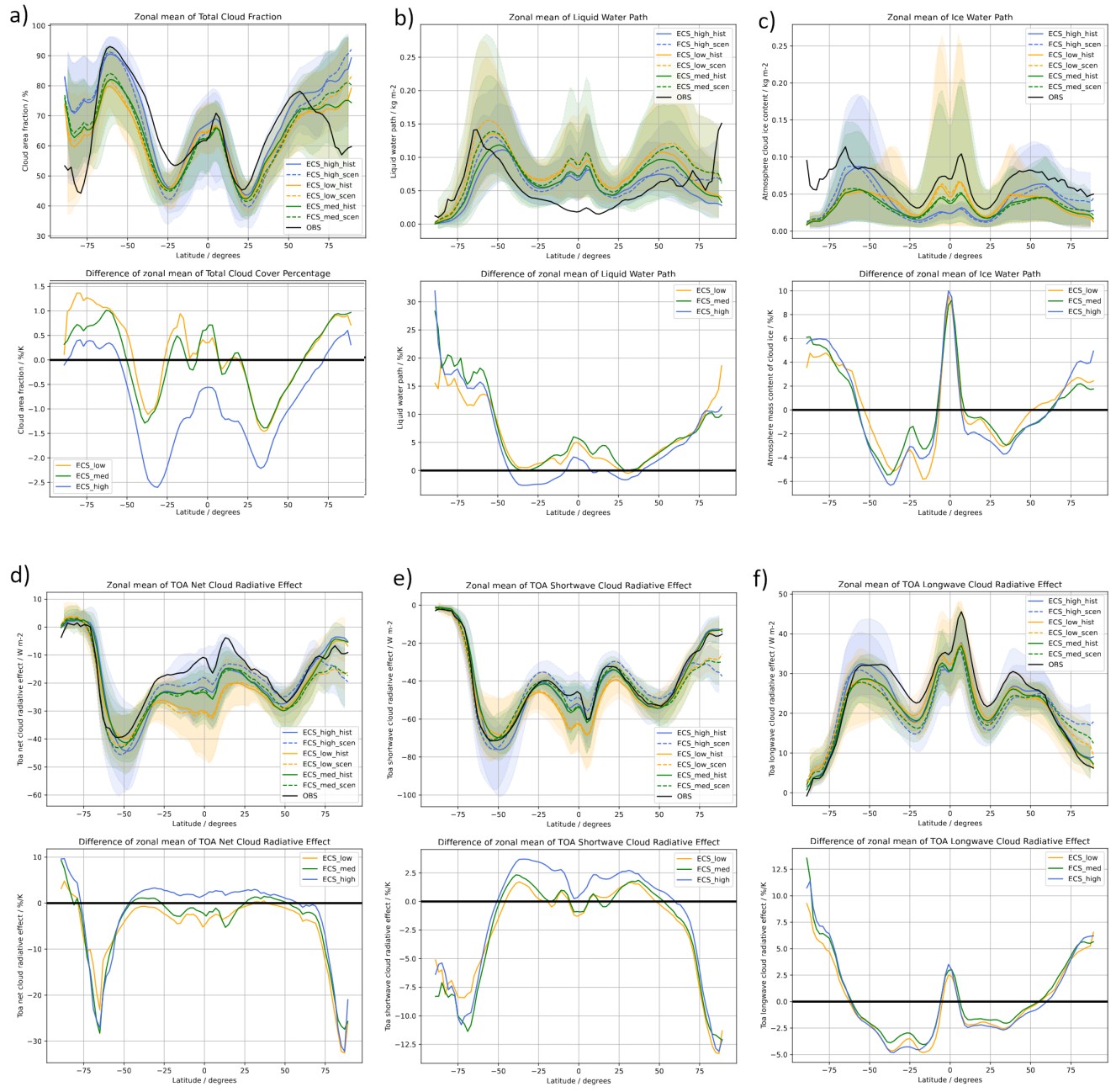

**Figure 9.** Upper panels: zonally averaged group means of (a) total cloud fraction, (b) liquid water path, (c) ice water path and (d) net, (e) shortwave and (f) longwave cloud radiative effect from historical simulations (solid lines) and RCP8.5 / SSP5-8.5 scenarios (dashed lines) for the three different ECS groups. The reference datasets are shown as solid black lines. Lower panels: corresponding relative differences of all zonally averaged group means between the RCP8.5 / SSP5-8.5 scenarios and the corresponding historical simulations. Shading indicates the 5% and 95% quantile of the single model results.

In all three subtropical stratocumulus regions investigated (North East Pacific, South East Pacific and South East Atlantic), the high ECS group shows a decrease in total cloud cover (Figure 11). In contrast, the low and medium ECS groups show particularly in the Southern Hemisphere stratocumulus regions an increase in total cloud cover that is most pronounced in the low ECS group.

In general, there is a decrease in cloud fraction in midlatitudes which is most pronounced in the high ECS group and becomes weaker towards the poles. In SH mid- and high latitudes south of 45°S, the low ECS group shows a strong positive sensitivity of up to more than 1%/K while the high ECS group shows a negative sensitivity of about -1%/K at 45°S. South of 55°S, the high ECS group also shows a positive sensitivity of total cloud cover. The medium ECS group lies in between the low and high ECS groups but is in general closer to the low ECS group. Averaged over the Southern Ocean (latitude belt 30-65°S), the high ECS models mostly show a negative sensitivity while the individual models in the two other groups show positive and negative sensitivities.

**Cloud liquid and ice water path**

In the Tropics between about 10°S and 10°N, the cloud ice water path shows a strong sensitivity to warming of up to 9%/K and 10%/K in all three ECS groups (Figure 9b). The zonally averaged ice water path increases also in all three groups north and south of about 60°N/S with the high ECS group showing the strongest sensitivity to warming. Particularly in the Arctic north of 80°N, the sensitivity of the simulated ice water path to warming is about twice as high in the high ECS group ( 4%/K) than in the medium and low ECS groups ( 2%/K). In midlatitudes, all groups show a negative sensitivity to warming with the high ECS group typically showing the strongest sensitivity in the Northern Hemisphere among the three ECS groups.

Similarly to the ice water path, also the zonally averaged liquid water path increases with temperature in all three groups in the polar regions (Figure 9c). This is consistent with the findings of Lelli et al. (2023) who report an observed trend to brighter and more liquid clouds in satellite measurements over the Arctic. In contrast to the ice water path, the lowest ECS group shows the highest sensitivity in the Arctic latitude belt. Averaged over the whole Arctic, however, there are no significant differences in ice and liquid water path over ocean between the different ECS groups (Figure 10a).

The amplitude of the decrease in ice water path per degree warming is peaking at about 35°S and N and is about twice as large in the Southern Hemisphere than in the Northern Hemisphere. Beyond about 60°N and S, there is an increase in ice water path that is getting more pronounced towards the poles. This increase in ice water path with warming is even stronger for the liquid water path with no significant differences between the ECS groups. This increase in liquid water path can be partly explained by a phase change from cloud ice to liquid at higher temperatures.

In the stratocumulus regions (Figure 11), liquid water path increases in the low ECS model group while it decreases in the high ECS group. The medium ECS group lies in between the two with many of the individual models disagreeing on the sign of the change. This behavior is consistent with the sensitivity of the changes in total cloud cover in these regions. We would like to note that ice water path values are typically very small in the stratocumulus regions. Relative changes can therefore be large without being physically relevant.

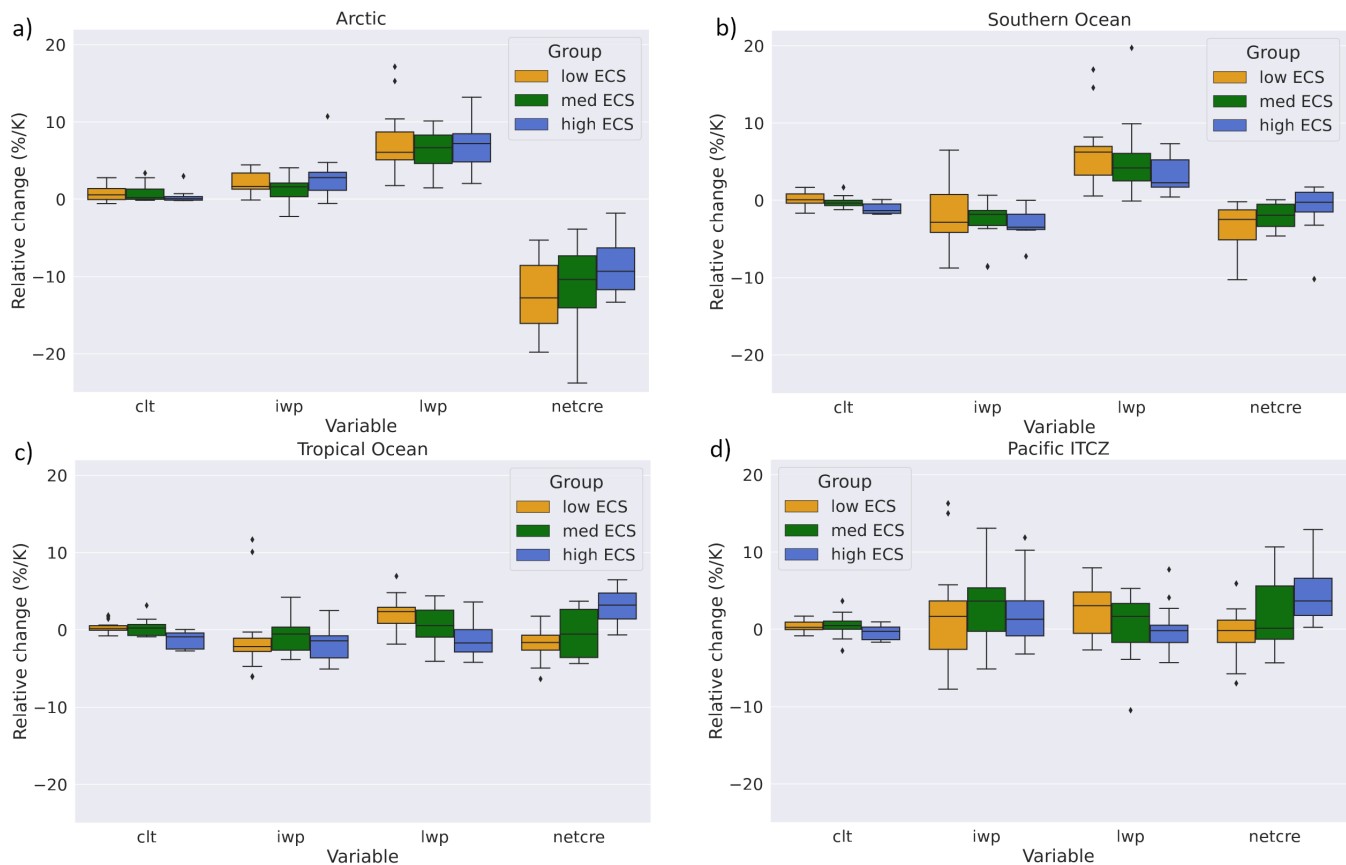

**Figure 10.** Relative change (calculated as the difference between the scenario value and the historical value divided by the historical value) of total cloud fraction (clt), ice water path (iwp), liquid water path (lwp) and net cloud radiative effect (netcre) per degree warming averaged over selected regions over the ocean: (a) Arctic (70-90°N), (b) Southern Ocean (30-65°S), (c) tropical ocean (30°N-30°S) and (d) Pacific ITCZ (0-12°N, 135°O-85°W). In the box plot, each box indicates the range from the first quartile to the third quartile, the vertical line shows the median and the whiskers the minimum and maximum values excluding the outliers. Outliers are defined as being outside 1.5 times the interquartile range.

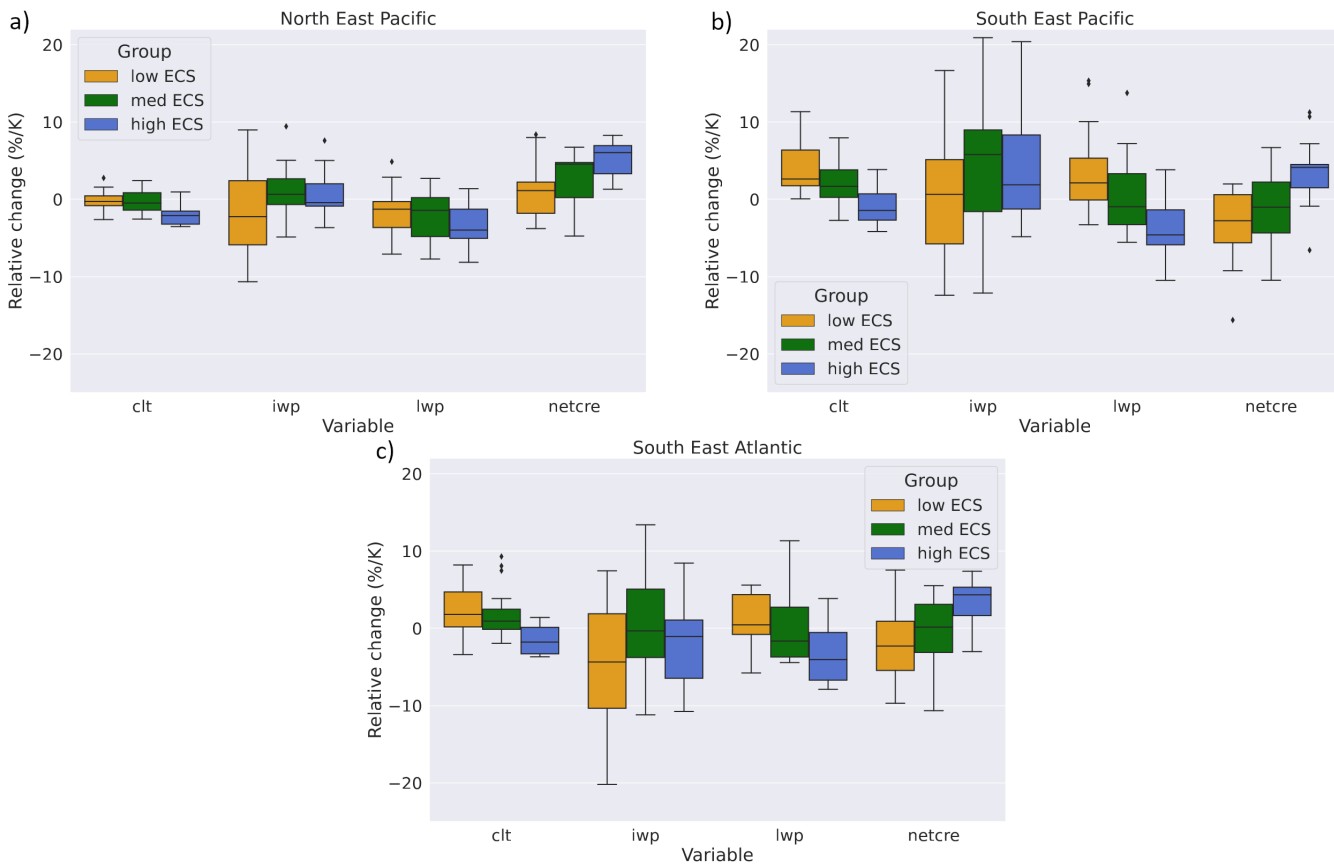

**Figure 11.** Same as Figure 10 for three stratocumulus regions (Muhlbauer et al., 2014), only over ocean: (a) North East Pacific (15-35°N, 120-140°W), (b) South East Pacific (10-30°S, 75-95°W) and (c) South East Atlantic (10-30°S, 10°W-10°O).

Over the Southern Ocean, the decrease in ice water path and the increase in liquid water path with warming is also not statistically significantly different among the three ECS groups. Averaged over the whole Southern Ocean (Figure 10b), all high ECS models show a decrease in cloud ice water path whereas about half of the low ECS models show an increase.

**Cloud radiative effects**

Over the northern polar region the cooling effect of the net cloud radiative effect increases significantly for all three ECS groups (Figure9def). Averaged over the whole Arctic (Figure 10a), the low ECS group shows the strongest increase in cooling among the three ECS groups. The increase in net cloud radiative effect is dominated by a stronger shortwave cloud radiative effect that is only partly compensated by a larger longwave cloud radiative effect. This is driven particularly by an increase in cloud liquid water path and only to a smaller extent to an increase in cloud ice water path and total cloud cover (Figures 9abc).

North of about 50°N and south of about 50°S, all three ECS groups show stronger shortwave cloud radiative effects, i.e. stronger cooling, in the future scenarios than in the historical simulations. In contrast, the shortwave cloud radiative effect is

335 reduced in the projections in mid- and low latitudes. Here, the low ECS group shows the smallest changes, while the reductions in shortwave cloud radiative effect per degree of warming are strongest in the high ECS group. This is mainly driven by a reduction in total cloud cover alongside a reduction in liquid water path that can only be compensated within about $\pm 10°$ around the Equator by an increase in cloud ice water path (Figure 9abc).

On average, there is a small decrease in the amplitude of the net radiative effect between about 1 and 3%/K for high ECS
models in the latitude belt 50°S to 50°N. For the two other groups there is a small increase in the amplitude. Beyond 50°N and 50°S the amplitude of the net cloud radiative effect increases (i.e. more negative) per degree temperature change with a peak at about 65°S and 80°N of about 25% and 30%, respectively, per degree temperature increase. Ceppi et al. (2016) shows that this cloud response results from an increasing cloud optical depth with temperature which is in agreement with the increased liquid water path in Figure 9c.

In the Tropics, the high ECS group shows the strongest weakening of the net cloud radiative effect. This is caused by a reduced shortwave cooling (Figure 9e) connected to the decrease in total cloud fraction. In contrast, the medium and low ECS groups show a stronger net cloud radiative effect (i.e. more negative) with warming in the future projections. This different behavior can also be seen in Figure 10c.

Driven mostly by the changes in total cloud cover and liquid water path, the cooling effect of the net cloud radiative effect in
the stratocumulus regions amplifies with warming in the low ECS group while it gets weaker in the high ECS group (Figure 11). Again, the medium ECS group is in between the two other groups with many individual models within this group disagreeing on the sign of the change in the net cloud radiative effect with warming.

## 4 Summary and conclusions

The uncertainty in the representation of clouds and their response to climate change is one of the main contributors to the
355 overall uncertainty in effective climate sensitivity and thus projections of future climate. The increased range of ECS obtained from the ensemble of CMIP6 models compared to previous CMIP phases motivated us to look into the differences in present-day and future projections of cloud parameters. For this, a total of 51 CMIP5 and CMIP6 models providing the required output were grouped by their ECS into three equally sized groups to investigate changes in cloud parameters in future projections from these models. Models with an ECS higher than 4.0 K belong to the "high" ECS group, with an ECS between 2.87 K and
360 4.0 K to the "medium" and with an ECS lower than 2.87 K to the "low" ECS group. Furthermore, historical simulations of the models were compared with satellite data to obtain a qualitative overview on the performance of the three model groups in simulating observed cloud patterns and properties.

Consistent with the findings of Kuma et al. (2023), we found that models with a high climate sensitivity typically have a better representation of observed cloud properties than models with a low or medium ECS. This is the case for most of
365 the variables investigated such as cloud ice water path, cloud radiative effects and precipitation. For fields that are already quite well simulated by CMIP models such as near-surface air temperature and TOA outgoing longwave radiation, only little differences were found among the three ECS groups.

The geographical pattern of total cloud cover simulated by the high ECS group is found to be in significantly better agreement with satellite observations than the two other group means. The global mean and RMSD from the high ECS group are smaller than the ones from the other groups, which tend to underestimate total cloud cover. Regarding cloud ice water path all group means underestimate the observed global mean and while at the same time they overestimate the global mean cloud liquid water path. As a result of the high observational uncertainty of global ice and liquid water path from satellite measurements, the model result for global mean ice and liquid water path are within the observational range (Lauer et al., 2023) making a quantitative assessment of the groups' performance difficult. The amplitude of the global mean cloud radiative effect is overestimated in the models with the largest bias found for the low ECS group. The geographical patterns of all model groups agree reasonably well with the observations. Again, the high ECS group shows the highest agreement among the three groups.

The better agreement of the geographical cloud patterns from the high ECS group with observations is particularly pronounced in midlatitudes (Southern Ocean and North Atlantic). Observed maxima in ice water path in midlatitudes and in particular over the Southern Ocean are best reproduced by this group. Other studies have already found that this could be related to an improved representation of supercooled liquid in some of the high ECS models (Tan et al., 2016; Zelinka et al., 2020). At the same time this model improvement leads to a decrease in the magnitude of the negative cloud phase feedback which results in a higher ECS (e.g., Bock et al., 2020; Zelinka et al., 2020; Bjordal et al., 2020; Frey and Kay, 2018). The liquid water path is overestimated in all models in the midlatitude stromtrack regions compared to the ESACCI Cloud dataset.

The observed local maxima in the amplitude of the net cloud radiative effect over the stratocumulus regions seen in observations are underestimated in all three group means and related to the known bias of underestimating the cloud fraction of stratocumulus clouds in the CMIP models (e.g., Jian et al., 2020).

In the Tropics the observed high ice water path values related to the ITCZ are underestimated by all three ECS groups with the low ECS group mean performing best. At the same time, the low ECS groups shows the highest overestimation of the net cloud radiative effect in the Tropics. The liquid water path in the ITCZ is overestimated by all models in respect to the ESACCI cloud dataset. We would like to note, however, that observational uncertainties of these quantities are quite large.

In order to investigate the sensitivity of cloud parameters to future warming simulated by the three ECS groups, we compared results from historical simulations with the ones from RCP8.5 and SSP5-8.5 runs from each group. We found that in polar regions, the increase in cloud cover per degree of warming is strongest in the low ECS models, which is about a factor of 2-3 higher than in the high ECS models. Together with an increase in cloud ice and liquid water path, the cooling effect of the net cloud radiative effect increases significantly for all three ECS groups particularly in the northern polar region. These simulated future changes in all three groups in polar regions are consistent with satellite observations showing an increase in the observed brightness of Arctic clouds in recent years (Lelli et al., 2023). Averaged over the whole Arctic, the low ECS group shows the strongest increase in the cooling effect of the shortwave cloud radiative effect among the three ECS groups.

In midlatitudes and in the Tropics, the three model groups do not agree on the sign of the sensitivity of cloud cover to warming. While the high ECS models show a decrease in cloud fraction particularly in SH mid- and high latitudes south of 45°S, the low ECS group shows a strong positive sensitivity of up to more than 1%/K. Over the tropical ocean, all high ECS models show a decrease in total cloud cover while the individual models in the two other groups do not agree on the sign of the

change. The shortwave cloud radiative effect is reduced in the projections in mid- and low latitudes with the low ECS group showing the smallest changes, while the reductions in shortwave cloud radiative effect per degree of warming are strongest in the high ECS group. This is mainly driven by a reduction in total cloud cover alongside a reduction in liquid water path that can only be compensated within about $\pm 10°$ around the Equator by an increase in cloud ice water path. Between about $10°S$ and $10°N$ all three ECS groups show a strong sensitivity of the cloud ice water path to warming of up to 9%/K and 10%/K. This increase in cloud ice water path is expected to be related to stronger and/or more frequent deep convection as the main increase in the vertical distribution of cloud ice occurs in the upper troposphere around 300 hPa and higher (not shown).

Similarly, the behavior of the three ECS groups is different in the subtropical stratocumulus regions. The high ECS group shows a decrease in total cloud cover with warming, the low and medium ECS groups show particularly in the SH stratocumulus regions an increase in total cloud cover. Together with changes in liquid water path following changes in cloud cover, the cooling effect of the net cloud radiative effect in the stratocumulus regions amplifies with warming in the low ECS group while it gets weaker in the high ECS group. Failure to reproduce observed trends in sea surface temperature gradient and therefore changes in inversion strength has found to be one possible reason for an overestimation of the positive cloud feedback in the stratocumulus regions (Cesana and Del Genio, 2021).

Over the Southern Ocean, we found a decrease in ice water path and an increase in liquid water path with warming. These changes, however, are not statistically significantly different among the three ECS groups. Averaged over the whole Southern Ocean (latitude belt 30-65°S), all high ECS models agree in a future decrease in cloud ice water path whereas about half of the low ECS models show a positive and half of the models a negative change in cloud ice.

Our results suggest that the differences in the net cloud radiative effect as a response to warming and thus differences in ECS among the CMIP models are not solely driven by an individual region but rather by changes in a range of cloud regimes leading to differences in the net cloud radiative effects. Contributors are changes in all different global cloud regimes, in polar regions, in tropical and subtropical regions and in midlatitudes. In polar regions, high ECS models show a significantly weaker increase in the net cooling effect of clouds due to warming than the low ECS models. At the same time, high ECS models show a decrease in the net cooling effect of clouds over the tropical ocean and the subtropical stratocumulus regions. In both regions low ECS models show either little change or even an increase in the cooling effect as a consequence of warming. The differences among the ECS groups in the Southern Ocean fit consistently into this picture, showing a higher sensitivity of the net cloud radiative effect to warming in the low ECS models than in the high ECS models. We thus conclude that changes in all three regions contribute to the amplitude of simulated ECS.

*Code and data availability.* All model simulations used for this paper are publicly available on ESGF. Observations used in the evaluation are detailed in Table 2. The observational datasets are not distributed with the ESMValTool that is restricted to the code as open source software. Observational datasets that are available through the Observations for Model Intercomparisons Project (obs4MIPs; https://esgf-node.llnl.gov/projects/obs4mips/) can be downloaded freely from the ESGF and directly used in the ESMValTool. For all other observational datasets, the ESMValTool provides a collection of scripts (NCL and Python) with exact downloading and processing instructions to recreate

the datasets used in this publication. All diagnostics used for this paper will be made available in the ESMValTool after acceptance of this publication. ESMValTool v2 is released under the Apache License, version 2.0. The latest release of ESMValTool v2 is publicly available on Zenodo at https://doi.org/10.5281/zenodo.3401363. The source code of the ESMValCore package, which is installed as a dependency of the ESMValTool v2, is also publicly available on Zenodo at https://doi.org/10.5281/zenodo.3387139. ESMValTool and ESMValCore are

developed on the GitHub repositories available at https://github.com/ESMValGroup with contributions from the community very welcome. For more information, we refer to the ESMValTool website (https://www.esmvaltool.org). All figures from this paper can be reproduced with the ESMValTool "recipe" (configuration script defining all datasets, processing steps and diagnostics to be applied) recipe_bock23acp.yml.

*Author contributions.* LB performed the analysis, prepared all figures, and lead the writing of the manuscript. AL contributed to the scientific interpretation of the results and the writing of the manuscript.

*Competing interests.* No competing interests.

*Acknowledgements.* This project has received funding from the European Union's Horizon 2020 research and innovation programme under Grant Agreement 101003536 (ESM2025—Earth System Models for the Future) and from the ESA Climate Change Initiative Climate Model User Group (ESA CCI CMUG).

We acknowledge the World Climate Research Programme, which, through its Working Group on Coupled Modelling, coordinated and

promoted CMIP6. We thank the climate modelling groups for producing and making available their model output, the Earth System Grid Federation (ESGF) for archiving the data and providing access, and the multiple funding agencies who support CMIP6 and ESGF. This work used resources of the Deutsches Klimarechenzentrum (DKRZ) granted by its Scientific Steering Committee (WLA) under project ID bd0854.

The ESA Climate Change Initiative (CCI) and Cloud_cci project are kindly acknowledged. The CERES-EBAF data were obtained from

the NASA Langley Research Center Atmospheric Science Data Center. The dataset MODIS used in this work was obtained from the obs4MIPs [https://esgf-node.llnl.gov/projects/obs4mips/ (accessed on 2 November 2021)] project hosted on the Earth System Grid Federation [https://esgf.llnl.gov (accessed on 2 November 2021)]. This manuscript contains modified Copernicus Climate Change Service (2021) information with ERA5 data retrieved from the Climate Data Store (neither the European Commission nor ECMWF is responsible for any use that may be made of the Copernicus information or data it contains). The NCEP-NCAR Reanalysis 1 data is provided by the NOAA

PSL, Boulder, Colorado, USA, from their website at https://psl.noaa.gov.

We thank Mattia Righi (DLR) for helpful comments on the manuscript.

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
