# Peer review of "Cloud properties and their projected changes in CMIP models with low/medium/high climate sensitivity"

_EGUsphere, 2023_

## Referee Comment (RC2)

**Review of "Cloud properties and their projected changes in CMIP models with low/medium/high climate sensitivity"**
**By Bock and Lauer**
**egusphere-2023-1086**

**Summary**
The authors compare climatological fields simulated by global climate models to those computed in observational datasets. The models are separated into high, medium, and low equilibrium climate sensitivity (ECS) categories, and it is found that for most fields examined, the high ECS models more closely resemble observations. Changes in these fields between the historical and future climate scenario are also examined, with the high ECS models showing largest changes in most location general. In general I was not impressed with this paper, for the reasons detailed below. I think the paper is flawed in its execution while also lacking a scientific motivation, and I therefore recommend rejection.

**Major Comments**
- The paper seems to lack any scientific question motivating the analysis or hypothesis that it is testing. Why are you evaluating these particular fields, and segregating the models by ECS? Is there a physical reason to expect the fidelity with which these fields match observations in the mean state to be tied to ECS? Do the authors believe that high ECS is more plausible than low ECS based on their results? What is the motivation for transitioning to examining how these fields change into the future? I didn't find any novel insights here that were not already well explained in the literature. In the end, I can't really understand what the point of the paper is, or why one would cite it.
- Most of the fields examined involve cloud properties (fractional area coverage, ice and liquid water path) or precipitation, but the evaluation is done without satellite simulators that ensure apples-to-apples comparisons of the geophysical fields (Bodas-Salcedo et al., 2011). It is well established within the community that one cannot simply compare a model cloud field to something retrieved from space, which has sampling biases, detection thresholds, scale differences, etc. Papers by Jen Kay, Greg Cesana, and others have made this point many times for several fields (G. Cesana & Chepfer, 2013; G. V. Cesana et al., 2021; J. E. Kay et al., 2012; Jennifer E. Kay et al., 2016, 2018). Even cloud radiative effect (clear- minus all-sky fluxes at the TOA) cannot be easily compared between models and observations because of differences in how clear-sky fluxes are provided in models vs observations (B. J. Sohn et al., 2010; B. J. Sohn & Bennartz, 2008; B.-J. Sohn et al., 2006). To facilitate more appropriate comparisons, adjusted clear-sky fluxes are now being provided by the CERES team (Loeb et al., 2020). For me, this decision to use raw model output to compare to satellite-retrieved fields is the most egregious flaw of the paper and I would need to see it remedied before I could recommend acceptance.
- I found it very disconcerting that the authors did not ensure a common time period for their model-observation comparisons. Why are the climatologies from the various observational products different from each other and from the models (1985-2004)?
- The Observations section was literally 3 sentences, none of which actually explained the datasets, their version/collection, nominal resolution, what instrument (on which satellite) is

measuring each geophysical quantity, etc.  This is unacceptable for a scientific manuscript in which models are being evaluated against observations.  The recurring cloud product with the acronym ESACCI is not even defined anywhere.

- In stark contrast to the 3-sentence Observations section, Section 2.3 reads like an advertisement for the ESMValTool.  Most of this information regarding the software you used to perform your analysis is meant for the *Code and data availability* section.

- The changes in cloud properties are computed by differencing the future scenario with the historical scenario.  While this will provide the total change in clouds, those changes will be due to an ambiguous mix of causes: responses to warming, decreases in aerosol loading, and adjustments from changes in other forcing agents.  High ECS models typically also have large aerosol-cloud interactions (Kiehl, 2007; Wang et al., 2021), so a portion of their change between historical and future climates will be due to a recovery from being strongly affected by aerosols in the historical period, and will not be purely attributable to cloud feedback processes.

**Specific Comments**
- Author list: both authors' names are in reverse order
- L7 and throughout: "both, cloud physical" the comma after both is not needed; this typo recurs throughout the paper (e.g., L26, 140)
- L99: what simulations are being used here? Also, it should be caveated that the change in cloud radiative effect is not the same as the cloud feedback owing to changes in clear-sky fluxes that are not related to clouds (Soden et al., 2004)
- L111: it doesn't matter which direction one is going; delete "when going from south to north"
- L111-113: these statements are made without providing any evidence of the role of changing cloud phase; suggest either deleting, citing the appropriate literature, or providing evidence.
- Figure 1: Given that ECS is strongly dependent on cloud feedback, it seems odd to plot cloud feedback on the y-axis, which is typically thought of as the dependent variable.
- L209: "clouds are warming" should be re-stated
- L226 vs L227: "largest positive bias"…."too strong net cloud radiative effect" – I'm confused about what these mean.  The net CRE is negative, so if it is "too strong" I'd expect that to mean that the negative magnitude is too large, but this would not be a positive bias.  Please restate.
- Figure 10 and elsewhere: I'm not sure what is meant be "relative change". How is this computed?
- Figure 11: if liquid water path is denoted as lwp rather than clwvi, it seems that ice water path should be denoted as iwp rather than clivi. Should one care about IWP over the stratocumulus regimes?
- L383-384: In this sentence, every possible regime on the planet is listed; is this really informative or helpful? If you quantified more rigorously the regimes that are strong contributors to inter-model spread in cloud feedback or ECS, you would find that not every location on the planet contributes equally.

**References**

Bodas-Salcedo, A., Webb, M. J., Bony, S., Chepfer, H., Dufresne, J. L., Klein, S. A., et al. (2011). COSP Satellite simulation software for model assessment. *Bulletin of the American Meteorological Society*, *92*(8), 1023–1043. https://doi.org/10.1175/2011bams2856.1

Cesana, G., & Chepfer, H. (2013). Evaluation of the cloud thermodynamic phase in a climate model using CALIPSO-GOCCP. *Journal of Geophysical Research: Atmospheres*, *118*(14), 7922–7937. https://doi.org/10.1002/jgrd.50376

Cesana, G. V., Ackerman, A. S., Fridlind, A. M., Silber, I., & Kelley, M. (2021). Snow Reconciles Observed and Simulated Phase Partitioning and Increases Cloud Feedback. *Geophysical Research Letters*, *48*(20), e2021GL094876. https://doi.org/10.1029/2021GL094876

Kay, J. E., Hillman, B. R., Klein, S. A., Zhang, Y., Medeiros, B., Pincus, R., et al. (2012). Exposing Global Cloud Biases in the Community Atmosphere Model (CAM) Using Satellite Observations and Their Corresponding Instrument Simulators. *Journal of Climate*, *25*(15), 5190–5207. https://doi.org/10.1175/JCLI-D-11-00469.1

Kay, Jennifer E., L'Ecuyer, T., Chepfer, H., Loeb, N., Morrison, A., & Cesana, G. (2016). Recent Advances in Arctic Cloud and Climate Research. *Current Climate Change Reports*, *2*(4), 159–169. https://doi.org/10.1007/s40641-016-0051-9

Kay, Jennifer E., L'Ecuyer, T., Pendergrass, A., Chepfer, H., Guzman, R., & Yettella, V. (2018). Scale-Aware and Definition-Aware Evaluation of Modeled Near-Surface Precipitation Frequency Using CloudSat Observations. *Journal of Geophysical Research: Atmospheres*, *123*(8), 4294–4309. https://doi.org/10.1002/2017JD028213

Kiehl, J. T. (2007). Twentieth century climate model response and climate sensitivity. *Geophys. Res. Lett.*, *34*. https://doi.org/10.1029/2007GL031383

Loeb, N. G., Rose, F. G., Kato, S., Rutan, D. A., Su, W., Wang, H., et al. (2020). Toward a Consistent Definition between Satellite and Model Clear-Sky Radiative Fluxes. *Journal of Climate*, *33*(1), 61–75. https://doi.org/10.1175/JCLI-D-19-0381.1

Soden, B. J., Broccoli, A. J., & Hemler, R. S. (2004). On the use of cloud forcing to estimate cloud feedback. *J. Climate*, *17*, 3661–3665. https://doi.org/10.1175/1520-0442(2004)0172.0.CO;2

Sohn, B. J., & Bennartz, R. (2008). Contribution of water vapor to observational estimates of longwave cloud radiative forcing. *J. Geophys. Res.*, *113*. https://doi.org/10.1029/2008JD010053

Sohn, B. J., Nakajima, T., Satoh, M., & Jang, H. S. (2010). Impact of different definitions of clear-sky flux on the determination of longwave cloud radiative forcing: NICAM simulation results. *Atmos. Chem. Phys.*, *10*(23), 11641–11646. https://doi.org/10.5194/acp-10-11641-2010

Sohn, B.-J., Schmetz, J., Stuhlmann, R., & Lee, J.-Y. (2006). Dry Bias in Satellite-Derived Clear-Sky Water Vapor and Its Contribution to Longwave Cloud Radiative Forcing. *Journal of Climate*, *19*(21), 5570–5580. https://doi.org/10.1175/JCLI3948.1

Wang, C., Soden, B. J., Yang, W., & Vecchi, G. A. (2021). Compensation Between Cloud Feedback and Aerosol-Cloud Interaction in CMIP6 Models. *Geophysical Research Letters*, *48*(4), e2020GL091024. https://doi.org/10.1029/2020GL091024

---

## Author Comment (AC1)

**Answer to RC1:**

Review of "Cloud properties and their projected changes in CMIP models with low/medium/high climate sensitivity"

by Lisa Bock and Axel Lauer

This paper presents an intercomparison of the simulation of clouds in the CMIP5 and CMIP6 multi-model ensembles. The models are grouped in three different categories: low, medium and high Effective Climate Sensitivity. In general, high-sensitivity models tend to perform better in the metrics analysed in this study. The paper is well written and the content is adequate for publication in ACP. It provides a valuable intercomparison, making it a useful addition to the scientific literature, and I recommend publication subject to minor revision. Please see my specific comments below.

**We thank Reviewer #1 for the constructive comments that helped improving the manuscript. We think we addressed all comments in the revised version and in our point-by-point answers below (given in bold). If not otherwise noted, all line numbers refer to the "track changes" version of the revised manuscript.**

GENERAL COMMENTS

I believe the results need to be put in the context of other intercomparisons that use different types of metrics. Studies like Brunner at al. (2020) reach very different conclusions by using a metrics that incorporate information about trends. I think this different approach needs to be critically discussed.

**Following the suggestion of the reviewer, we added a paragraph about relevant studies using different approaches and metrics.**
**Lines 59-64: "The performance of CMIP models has also been investigated in other studies. For example, Kuma et al. (2023) applied an artificial neural network to derive cloud types from radiation fields. They found that results from models with a high ECS agree on average better with observations than from models with a low ECS. Jiang et al. (2021) found that the models' ECS is positively correlated with the integrated cloud water content and water vapor performance scores for both CMIP6 and CMIP5 models. In contrast, Brunner et al. (2020) showed that some CMIP6 models with high future warming compared to other models receive systematically lower performance weights when using anomaly, variance, and trend of surface air temperature, and anomaly and variance of sea level pressure to assess the models' performance. "**

A more detailed description of the caveats in the comparisons of the IWP is needed. The model variable used (clivi) includes precipitating frozen hydrometeors only if the precipitating hydrometeor is seen by the model's radiation code. This is model-dependent and can introduce significant biases in the comparisons. Also, I wonder if the observational datasets chosen are representative of the diversity in observational estimates. Both ESACCI and MODIS are based on passive retrievals, and therefore will share similar caveats and biases (Waliser et al., 2009). I'd suggest using an alternative reference dataset based on a different remote sensing technology like CloudSat.

**Thanks for pointing that out. We changed the alternative measurement of iwp and lwp in Fig. 3 for the pattern correlation to the CloudSat dataset. We used the same version as Lauer et al. (2023), who excluded precipitating columns to estimate cloud water path values with no precipitating particles.**

**To highlight the observational uncertainties, we added:**
**Lines 175-180: "For cloud ice and cloud liquid water path the pattern correlations between ESACCI Cloud (passive instrument) and the alternative measurements of CloudSat (active instrument) show the large uncertainties of these quantities derived from satellite observations (e.g., Lauer et al., 2023). An additional uncertainty in this comparison is introduced, as some CMIP models may provide the sum of cloud ice and falling ice (e.g. snow) in the ice water path values if the falling ice is included in their radiation calculations. The number of models including falling ice radiative effects, however, is rather small and thus not expected to play an important role in the group means. An overview can be found e.g. in Li et al. (2020a), their Table 1."**

SPECIFIC COMMENTS

-L41-42: there are other studies that looked into the reasons for the increased in sensitivity in specific models, like Gettelman et al. (2019) and Bodas-Salcedo et al. (2019). It's worth noting that coupled feedbacks (e.g. sea-ice albedo) can play a significant role in some models (Andrews et al., 2019).

**As suggested, we added these citations and now mention also connections to other coupled feedbacks.**
**Lines 47-49: "They also point out that the simulated present-day mean state of cloud properties is correlated with the simulated cloud feedback but could also be connected to other coupled feedbacks (Andrews et al., 2019)."**

- Table 2. Please specify which CERES-EBAF version you've used. Also, the reference for ERA5 is missing.

**We added the version of CERES-EBAF (Ed4.2) to Table 2 and fixed the citation entry of ERA5.**

REFERENCES

Andrews, T., Andrews, M. B., Bodas-Salcedo, A., Jones, G. S., Kuhlbrodt, T., Manners, J., et al. (2019). Forcings, feedbacks, and climate sensitivity in HadGEM3-GC3.1 and UKESM1. Journal of Advances in Modeling Earth Systems, 11, 4377– 4394. https://doi.org/10.1029/2019MS001866.

Bodas-Salcedo, A., Mulcahy, J. P., Andrews, T., Williams, K. D., Ringer, M. A., Field, P. R., & Elsaesser, G. S. (2019). Strong dependence of atmospheric feedbacks on mixed-phase microphysics and aerosol-cloud interactions in HadGEM3. Journal of Advances in Modeling Earth Systems, 11, 1735– 1758. https://doi.org/10.1029/2019MS001688.

Brunner, L., Pendergrass, A. G., Lehner, F., Merrifield, A. L., Lorenz, R., and Knutti, R. (2020): Reduced global warming from CMIP6 projections when weighting models by performance and independence, Earth Syst. Dynam., 11, 995–1012, https://doi.org/10.5194/esd-11-995-2020.

Gettelman, A., Hannay, C., Bacmeister, J. T., Neale, R. B., Pendergrass, A. G., Danabasoglu, G., et al. (2019). High climate sensitivity in the Community Earth System Model Version 2 (CESM2). Geophysical Research Letters, 46, 8329– 8337, https://doi.org/10.1029/2019GL083978.

Kuma, P., Bender, F. A.-M., Schuddeboom, A., McDonald, A. J., and Seland, O. (2023): Machine learning of cloud types in satellite observations and climate models, Atmospheric Chemistry and Physics, 23, 523–549, https://doi.org/10.5194/acp-23-523-2023.

Lauer, A., Bock, L., Hassler, B., Schröder, M., and Stengel, M. (2023): Cloud Climatologies from Global Climate Models – A Comparison of CMIP5 and CMIP6 Models with Satellite Data, Journal of Climate, 36, 281–311, https://doi.org/10.1175/JCLI-D-22-0181.1.

Li, J.-L. F., Xu, K.-M., Jiang, J. H., Lee, W.-L., Wang, L.-C., Yu, J.-Y., Stephens, G., Fetzer, E., and Wang, Y.-H. (2020): An Overview of CMIP5 and CMIP6 Simulated Cloud Ice, Radiation Fields, Surface Wind Stress, Sea Surface Temperatures, and Precipitation Over Tropical and Subtropical Oceans, Journal of Geophysical Research: Atmospheres, 125, e2020JD032 848, https://doi.org/10.1029/2020JD032848.

Waliser, D., et al. (2009), Cloud ice: A climate model challenge with signs and expectations of progress, J. Geophys. Res., 114, D00A21, doi:10.1029/2008JD010015.

---

## Author Comment (AC2)

**Answer to RC2:**

Review of "Cloud properties and their projected changes in CMIP models with low/medium/high climate sensitivity"
By Bock and Lauer
egusphere-2023-1086

Summary

The authors compare climatological fields simulated by global climate models to those computed in observational datasets. The models are separated into high, medium, and low equilibrium climate sensitivity (ECS) categories, and it is found that for most fields examined, the high ECS models more closely resemble observations. Changes in these fields between the historical and future climate scenario are also examined, with the high ECS models showing largest changes in most location general. In general I was not impressed with this paper, for the reasons detailed below. I think the paper is flawed in its execution while also lacking a scientific motivation, and I therefore recommend rejection.

**We thank Reviewer #2 for helping to improve the paper. All points raised by the reviewer are well taken. We think we addressed all concerns in the revised version and in our point-by-point answers below (given in bold). If not otherwise noted, all line numbers refer to the "track changes" version of the revised manuscript.**

Major Comments

- The paper seems to lack any scientific question motivating the analysis or hypothesis that it is testing. Why are you evaluating these particular fields, and segregating the models by ECS? Is there a physical reason to expect the fidelity with which these fields match observations in the mean state to be tied to ECS? Do the authors believe that high ECS is more plausible than low ECS based on their results? What is the motivation for transitioning to examining how these fields change into the future? I didn't find any novel insights here that were not already well explained in the literature. In the end, I can't really understand what the point of the paper is, or why one would cite it.

  **The discussions about high ECS values from some of the CMIP6 models and the well-known large contribution of cloud feedbacks to the uncertainty range of ECS (e.g. Kuma et al., 2023; Jiang et al., 2021) motivated us to look into differences in simulated physical cloud properties and cloud radiative effects between future and present-day simulations. Of particular interest was whether there are systematic differences in these cloud-related quantities between different groups of models categorized by their ECS. Here, in particular the sensitivity of the physical properties to warming is of interest, as those give some insight into the uncertainty of the projected cloud properties and their potential contribution to cloud feedbacks and ECS. An assessment of the present-day model performance beyond a qualitative analysis to help interpreting simulated future changes in cloud properties or an assessment of differences in the plausibility of certain ECS values is not part of this study. We are not aware of any other study looking into simulated future changes in physical cloud properties from CMIP models. As the models are quite different in their sensitivity to the prescribed forcings, it makes sense to group models by certain**

**characteristics to facilitate analysis and obtain more general conclusions beyond individual models. As future changes in cloud properties are closely connected to cloud feedbacks and cloud feedbacks are strongly correlated with ECS, we use ECS as a simple proxy to group the ensemble of CMIP5 and CMIP6 models. To our knowledge, this is a novel approach and has not been published before.**

**We rephrased and clarified our motivation and our approach in the revised version, references to previous work have been extended.**
**We added to the abstract: "ECS is used as a simple metric to group the models as the sensitivity of the physical cloud properties to warming is closely related to cloud feedbacks, which in turn are known to have a large contribution to ECS." (Lines 8-10) and "In order to help interpreting the projected changes, model results from historical simulations are also compared to observations." (Lines 12-13)**
**We added a paragraph to the introduction: "As future changes in cloud properties are closely connected to cloud feedbacks and cloud feedbacks are known to be strongly correlated with ECS (see Sect. 3.1), we use ECS as a simple proxy to group the ensemble of CMIP5 and CMIP6 models for this analysis. This facilitates the analysis and allows for obtaining more general conclusions beyond individual models that can vary widely in their sensitivity to the prescribed forcings. A particular focus of this study is whether there are systematic differences in cloud-related quantities between the different ECS groups. The sensitivity of the physical properties to warming is analysed, as this gives some insight into the uncertainty of the projected cloud properties and their potential contribution to cloud feedbacks and ECS." (Lines 50-56)**
**We added to the summary: "Furthermore, historical simulations of the models were compared with satellite data to obtain a qualitative overview on the performance of the three model groups in simulating observed cloud patterns and properties." (Lines 380-382)**
**We also clarified that an assessment of the present-day model performance beyond a qualitative analysis to help interpreting simulated future changes in cloud properties or an assessment of differences in the plausibility of certain ECS values is not part of this study: "A qualitative assessment of the present-day model performance by comparing key cloud properties with satellite data is done to help interpreting simulated future changes in cloud properties. We would like to note that conclusions on the plausibility of certain ECS values cannot be drawn from this comparison and are thus not an aim of this study." (Lines 57-59).**

- Most of the fields examined involve cloud properties (fractional area coverage, ice and liquid water path) or precipitation, but the evaluation is done without satellite simulators that ensure apples-to-apples comparisons of the geophysical fields (Bodas-Salcedo et al., 2011). It is well established within the community that one cannot simply compare a model cloud field to something retrieved from space, which has sampling biases, detection thresholds, scale differences, etc. Papers by Jen Kay, Greg Cesana, and others have made this point many times for several fields (G. Cesana & Chepfer, 2013; G. V. Cesana et al., 2021; J. E. Kay et al., 2012; Jennifer E. Kay et al., 2016, 2018). Even cloud radiative effect (clear- minus all-sky fluxes at the TOA) cannot be easily compared between models and observations because of differences in how clear-sky fluxes are provided in models vs observations (B. J. Sohn et al., 2010; B. J. Sohn & Bennartz, 2008; B.-J. Sohn et al., 2006). To facilitate more appropriate comparisons, adjusted clear-sky fluxes are now being provided by the CERES team (Loeb et al., 2020). For me, this decision to use raw

model output to compare to satellite-retrieved fields is the most egregious flaw of the paper and I would need to see it remedied before I could recommend acceptance.

**We agree with the reviewer that a comparison of model output generated by satellite simulators with the satellite observations would be more appropriate. In CMIP5 and CMIP6, however, such output is only available from a very limited number of models (historical simulation: 15 CMIP6 models + 10 CMIP5 models; scenario simulation: 7 CMIP6 models + 2 CMIP5 models). In addition, of all variables investigated here, only total cloud fraction is available, other variables from satellite simulators such as liquid and ice water path or radiative fluxes are not available for these models. This study would therefore not be possible with the available CMIP model output generated by satellite simulators.**
**The main aim of this work to investigate whether there are systematic differences in present-day and projected cloud properties between different groups of models sorted by their ECS and if so, to quantify and document these differences. The comparison of the models with satellite observations does not aim at a quantitative evaluation of the different model groups or assessing whether high ECS values are more realistic than low ECS values but rather to help interpreting the projected changes in cloud properties.**

**We clarified this in the revised version and emphasized the qualitative aspect of this comparison. The limitations of this comparison because of the lack of output from satellite simulators has been extended and highlighted: "Most of the CMIP5 and CMIP6 historical simulations, however, do not provide such output. Of all variables investigated here, only total cloud fraction is available, other variables from satellite simulators such as cloud liquid and ice water path or radiative fluxes are not available for these models. … We would like to note that this limitation restricts a quantitative assessment of differences between models and observations as an unknown error is introduced by comparing two not fully equal quantities regarding their definition (e.g. observational thresholds) as well as temporal and spatial sampling. An assessment of the present-day model performance beyond a qualitative analysis to help interpreting simulated future changes in cloud properties or an assessment of differences in the plausibility of certain ECS values is therefore not possible." (Lines 191-201)**

**Regarding the evaluation of the cloud radiative effect we updated the reference dataset CERES EBAF from version Ed2.7 to Ed4.2. Starting at v4.1, the dataset provides adjusted clear-sky fluxes which are now defined in a manner that is more in line with how clear-sky fluxes are represented in climate models. Figure 3, Table 4, Figure 7 and 9 have been updated accordingly.**
**There are differences in the net cloud radiative forcing using v4.2 instead of v2.7 which are largest in the Tropics. The decrease of the netcre in the Tropics results from a larger lwcre (more warming) and a weaker swcre (less cooling) in the newer version. As we compare the different model groups in first place, these changes have no influence on the conclusions made in the paper.**

- I found it very disconcerting that the authors did not ensure a common time period for their model-observation comparisons. Why are the climatologies from the various observational products different from each other and from the models (1985-2004)?

**We choose the time period 1985-2004 for the climate models as this period is covered**

by both, the CMIP5 and CMIP6 historical model runs making it consistent to include both model generations in each ECS group. The time period of the reference datasets depends on the data availability for the specific reference dataset. While this choice of model years is somewhat arbitrary and does not match the years of the observations exactly, we found that it has very little impact on the multi-year group averages. This is not surprising as ESMs are not expected to reproduce the exact observed phase of climate modes largely controlling present-day variability of clouds but rather their statistical properties.
This has been clarified in the revised version by adding: "We would like to note that the time period from the models used for comparison with the observations (see Sect. 2.1) does not match exactly the observed years. It is not surprising, however, that this has very little impact on the multi-year group averages as ESMs are not expected to reproduce the exact observed phase of climate modes largely controlling present-day variability of clouds but rather their statistical properties." (Lines 96-99)

- The Observations section was literally 3 sentences, none of which actually explained the datasets, their version/collection, nominal resolution, what instrument (on which satellite) is measuring each geophysical quantity, etc. This is unacceptable for a scientific manuscript in which models are being evaluated against observations. The recurring cloud product with the acronym ESACCI is not even defined anywhere.

  Following the reviewer's comment, we extended the observations section now giving a brief description of all datasets used, including references for more detailed information (Lines 100-114). We prefer to keep these descriptions short in order to not simply repeat other studies and lengthen the paper unnecessarily.

- In stark contrast to the 3-sentence Observations section, Section 2.3 reads like an advertisement for the ESMValTool. Most of this information regarding the software you used to perform your analysis is meant for the Code and data availability section.

  We shorten the section as suggested now mostly referring to other studies for details on ESMValTool.

- The changes in cloud properties are computed by differencing the future scenario with the historical scenario. While this will provide the total change in clouds, those changes will be due to an ambiguous mix of causes: responses to warming, decreases in aerosol loading, and adjustments from changes in other forcing agents. High ECS models typically also have large aerosol-cloud interactions (Kiehl, 2007; Wang et al., 2021), so a portion of their change between historical and future climates will be due to a recovery from being strongly affected by aerosols in the historical period, and will not be purely attributable to cloud feedback processes.

  In this paper we investigate whether there are systematic differences in simulated cloud properties between the three ECS groups. The aim is to quantify and document such differences. For a better comparability of the different model groups, we also calculate the change in cloud properties per degree of surface warming. An attribution, however, to differences in cloud feedbacks is beyond the scope of this paper for the reasons given by the reviewer and therefore not done in this study.

  We corrected "cloud feedback" with "cloud radiative effect" in Line 18 and Line

**444.**

Specific Comments

- Author list: both authors' names are in reverse order

  **Thanks for spotting. We fixed that.**

- L7 and throughout: "both, cloud physical" the comma after both is not needed; this typo recurs throughout the paper (e.g., L26, 140)

  **Thank you for pointing this out. We deleted all commas after "both".**

- L99: what simulations are being used here? Also, it should be caveated that the change in cloud radiative effect is not the same as the cloud feedback owing to changes in clear-sky fluxes that are not related to clouds (Soden et al., 2004)

  **In order to calculate the cloud feedback, we use the simulations forced by an abrupt quadrupling of $CO_2$ (abrupt-4 × CO2) and the preindustrial control simulations (piControl). The same model simulations are used to obtain ECS. We clarified this in the revised text.**
  **We would like to stress, that we do not investigate cloud feedbacks beyond our motivation that ECS is correlated with cloud feedbacks. Quantitative analyses of cloud feedbacks are not performed is study.**

  **Line 84: We added "…and cloud feedbacks…".**
  **Lines 134/135: We added "…from abrupt-4×CO2 simulations compared to the corresponding piControl simulations."**

- L111: it doesn't matter which direction one is going; delete "when going from south to north"

  **Removed as suggested.**

- L111-113: these statements are made without providing any evidence of the role of changing cloud phase; suggest either deleting, citing the appropriate literature, or providing evidence.

  **We added the reference Ceppi et al. (2017) as citation for this statement (Line 153).**

- Figure 1: Given that ECS is strongly dependent on cloud feedback, it seems odd to plot cloud feedback on the y-axis, which is typically thought of as the dependent variable.

  **We modified Figure 1 as suggested by swapping the x- with y-axis.**

- L209: "clouds are warming" should be re-stated

  **We rephrased the sentence to: "Clouds have a warming radiative …" (Lines 263).**

- L226 vs L227: "largest positive bias" ..." too strong net cloud radiative effect" – I'm confused about what these mean. The net CRE is negative, so if it is "too strong" I'd

expect that to mean that the negative magnitude is too large, but this would not be a positive bias. Please restate.

**The "largest positive bias" could be found over the stratocumulus regions, which means that the negative cloud radiative effect is too weak in the models over these regions. Beside these regions there is a "too strong net cloud radiative effect" in the Tropics between 30°N and 30°S. This means the netCRE is more negative in the models than in the observations. So the two statements mentioned by the reviewer do not belong together. We clarified this in the revised manuscript by adding "Apart from that, …" (Lines 281/282) in the beginning of the second sentence.**

- Figure 10 and elsewhere: I'm not sure what is meant be "relative change". How is this computed?

  **Thanks for spotting that. We added the definition to the text and the caption of Figure 10.**

  **Lines 294/295: "The relative changes (calculated as the differences between the scenario value and the historical value divided by the historical value)…"**

- Figure 11: if liquid water path is denoted as lwp rather than clwvi, it seems that ice water path should be denoted as iwp rather than clivi. Should one care about IWP over the stratocumulus regimes?

  **We replaced "clivi" with "iwp" in the text as suggested as well as in the Figures 5, 10 and 11.**
  **Please note that "clwvi" is defined as the sum of the cloud liquid and ice water path. Liquid water path is calculated as lwp = clwvi - clivi.**

  **We show iwp in the Figure 11 for completeness. As iwp values are close to zero in the stratocumulus regions, relative changes can be large but are physically not relevant. This has been added to the text: "We would like to note that ice water path values are typically very small in the stratocumulus regions. Relative changes can therefore be large without being physically relevant." (Lines 340-342)**

- L383-384: In this sentence, every possible regime on the planet is listed; is this really informative or helpful? If you quantified more rigorously the regimes that are strong contributors to inter-model spread in cloud feedback or ECS, you would find that not every location on the planet contributes equally.

  **The reviewer is right, our results show that not a single but rather all cloud regimes are important. We clarified this in the revised manuscript by extending the corresponding sentence (Line 446): "…in all different global cloud regimes, …"**

References

Bodas-Salcedo, A., Webb, M. J., Bony, S., Chepfer, H., Dufresne, J. L., Klein, S. A., et al. (2011). COSP Satellite simulation software for model assessment. Bulletin of the American Meteorological Society, 92(8), 1023–1043. https://doi.org/10.1175/2011bams2856.1

Ceppi, P., Brient, F., Zelinka, M.D. and Hartmann, D.L. (2017), Cloud feedback mechanisms and their representation in global climate models. WIREs Clim Change, 8: e465. https://doi.org/10.1002/wcc.465

Cesana, G., & Chepfer, H. (2013). Evaluation of the cloud thermodynamic phase in a climate model using CALIPSO-GOCCP. Journal of Geophysical Research: Atmospheres, 118(14), 7922–7937. https://doi.org/10.1002/jgrd.50376

Cesana, G. V., Ackerman, A. S., Fridlind, A. M., Silber, I., & Kelley, M. (2021). Snow Reconciles Observed and Simulated Phase Partitioning and Increases Cloud Feedback. Geophysical Research Letters, 48(20), e2021GL0948ti6. https://doi.org/10.1029/2021GL094876

Jiang, J. H., Su, H., Wu, L., Zhai, C., and Schiro, K. A. (2021): Improvements in Cloud and Water Vapor Simulations Over the Tropical Oceans in CMIP6 Compared to CMIP5, Earth and Space Science, 8, e2020EA001 520, https://doi.org/https://doi.org/10.1029/2020EA001520

Kay, J. E., Hillman, B. R., Klein, S. A., Zhang, Y., Medeiros, B., Pincus, R., et al. (2012). Exposing Global Cloud Biases in the Community Atmosphere Model (CAM) Using Satellite Observations and Their Corresponding Instrument Simulators. Journal of Climate, 25(15), 5190–5207. https://doi.org/10.1175/JCLI-D-11-00469.1

Kay, Jennifer E., L'Ecuyer, T., Chepfer, H., Loeb, N., Morrison, A., & Cesana, G. (2016). Recent Advances in Arctic Cloud and Climate Research. Current Climate Change Reports, 2(4), 159–169. https://doi.org/10.1007/s40641-016-0051-9

Kay, Jennifer E., L'Ecuyer, T., Pendergrass, A., Chepfer, H., Guzman, R., & YePella, V. (2018). Scale-Aware and Definition-Aware Evaluation of Modeled Near-Surface Precipitation Frequency Using CloudSat Observations. Journal of Geophysical Research: Atmospheres, 123(8), 4294–4309. https://doi.org/10.1002/2017JD028213

Kiehl, J. T. (2007). Twentieth century climate model response and climate sensitivity. Geophys. Res. Lett., 34. https://doi.org/10.1029/2007GL031383

Kuma, P., Bender, F. A.-M., Schuddeboom, A., McDonald, A. J., and Seland, O. (2023): Machine learning of cloud types in satellite observations and climate models, Atmospheric Chemistry and Physics, 23, 523–549, https://doi.org/10.5194/acp-23-523-2023

Loeb, N. G., Rose, F. G., Kato, S., Rutan, D. A., Su, W., Wang, H., et al. (2020). Toward a Consistent Definition between Satellite and Model Clear-Sky Radiative Fluxes. Journal of Climate, 33(1), 61–75. https://doi.org/10.1175/JCLI-D-19-0381.1

Soden, B. J., Broccoli, A. J., & Hemler, R. S. (2004). On the use of cloud forcing to estimate cloud feedback. J. Climate, 17, 3661–3665. https://doi.org/10.1175/1520-0442(2004)017<3661:OTUOCF>2.0.CO;2

Sohn, B. J., & Bennartz, R. (2008). Contribution of water vapor to observational estimates of longwave cloud radiative forcing. J. Geophys. Res., 113. https://doi.org/10.1029/2008JD010053

Sohn, B. J., Nakajima, T., Satoh, M., & Jang, H. S. (2010). Impact of different definitions of clear-sky flux on the determination of longwave cloud radiative forcing: NICAM simulation

results. Atmos. Chem. Phys., 10(23), 11641–11646. https://doi.org/10.5194/acp-10-11641-2010

Sohn, B.-J., Schmetz, J., Stuhlmann, R., & Lee, J.-Y. (2006). Dry Bias in Satellite-Derived Clear-Sky Water Vapor and Its Contribution to Longwave Cloud Radiative Forcing. Journal of Climate, 19(21), 5570–5580. https://doi.org/10.1175/JCLI3948.1

Wang, C., Soden, B. J., Yang, W., & Vecchi, G. A. (2021). Compensation Between Cloud Feedback and Aerosol-Cloud Interaction in CMIP6 Models. Geophysical Research Letters, 48(4), e2020GL091024. https://doi.org/10.1029/2020GL091024

---

## Referee Report (RR1)

Review of "Cloud properties and their projected changes in CMIP models with low/medium/high climate sensitivity" by L. Bock and A. Lauer.

In this study, the authors classify outputs of 51 CMIP5 and CMIP6 models into low, medium and high ECS groups and then compare them with observations. They further look at the change in cloud properties between historical and 4xCO2 simulations weighted by global mean surface warming. They find that the models from the high ECS group better represent the observed climatology of cloud-related variables and have different sensitivities to warming than the low- and medium-ECS groups.

The topic of this paper aligns well with the scope of the journal. While I recognize the value of investigating the response of clouds to warming and the amount of work it takes in terms of data processing, I find that the study suffers from two major flaws: a non-consistent direct comparison of model cloud-related outputs with observations and the use of coupled historical simulations in the model evaluation. Also, it seems that the authors have already evaluated these models in a separate paper, so I see little value in doing this again. However, I find the analysis of the cloud response to climate change very interesting. More details are given below.

Main comments:
My biggest concern is the comparison of cloud-related fields with observations, which doesn't account for observational uncertainties and inherent limitations of the satellite instruments. The LWP products suffer from large uncertainties (sometimes several times greater than the observed value itself, Lebsock and Su, 2014; Elsaesser et al., 2017) and cannot be used to assess models on a global scale. IWP products seem to more reliable but there is still the question of whether precipitation is accounted for or not (e.g., Li et al., 2014). The cloud fraction also cannot be compared directly to observations because of the instrument limitations and the difference in cloud definitions between models and observations. I'm attaching a figure showing the impact of using ISCCP (basically AVHRR), MODIS and CALIPSO simulators on the original output of the model for 3 CMIP6 models. The differences are very large, region dependent and model dependent…

[Figure]

*Figure 1: Effect of ISCCP, MODIS and CALIPSO simulators on total cloud cover in three CMIP6 models. Total cloud cover ('clt') as simulated by CNRM-CM6, GFDL-CM4 and CanESM5 CMIP6 models (first row) and the difference between the original total cloud cover and that simulated by ISCCP (second row), MODIS (third row) and CALIPSO (fourth row) simulators.*

The second main concern is the comparison of historical simulations with present-day observations that have not the same surface forcings. The SST pattern and magnitude, which have strong impact on all the

variables that are studied here, are not well reproduced by the coupled models as shown in the literature (e.g., Seager et al., 2019). Therefore, it is not a fair comparison. Instead, the authors should use AMIP type simulations to assess the models.

Another main comment, which could be easily fixed, is the conclusion. Except for the last paragraph, which is very insightful, the conclusion is far too long (2 pages) and does not summarize the results but rather re-state them without any apparent structure.

Minor comments:
Almost no information about the observations used is given and including potential uncertainties, which are raised here and there in the manuscript but without being formally quantified. As is, it looks like the authors have very little knowledge about the observations they're using.

I couldn't find a clear definition of how the feedbacks are computed.

The introduction is not doing the best job at motivating the ECS discrimination. If the idea is that larger cloud feedback could be related to mean state cloud properties, then I would classify the models by the global mean feedback rather than ECS, because the ECS is affected by other feedbacks than those from clouds. The way it is presented in the paper is even slightly backward in my opinion.

I question the usefulness of having 2 to 3 versions of a model with the same atmospheric component especially when it comes to evaluating atmospheric quantities. They have disproportional impact on the mean. This question is not specific to that study though.

L129-130: then why is there no distinction between CMIP5 and CMIP6 models?

L136-139: I'm not sure what this means. There can be ice clouds at all latitudes in the high levels. This latitude loosely corresponds to where clouds can be mixed-phase cloud almost year-round. These clouds show different feedbacks from the warm clouds.

L158: simulated is written two times in a row and I don't think that Z20 is saying or showing such a conclusion in their study.

Fig. 3: Aside from the non-consistent model-to-observations comparison, the spread and SD appear to be very similar between the group, so the differences are not significant and there is no clear systematic behavior among the groups.

L198-199: I don't understand this sentence.

L243: not the number on the figure, so I suppose the authors decided to switch the default dataset to CERES during the first round of review, but failed to revise the text.

I don't see the added value of Fig. 10 and 11 compared to Fig. 9. I think that Fig. 9 and the analysis associated to it is the best part of this manuscript and should be the focus of this study.

Each main Sc decks is singled out yet no there is no motivation for doing this, I'm also not a fan of picking fixed regions to study cloud response to warming since these decks can evolve in terms of location.

The beginning of the conclusion is confusing. The authors argue that the increase of ECS between CMIP generation motivated them to investigate the response of clouds to climate change, yet most of the study is based on present-day climate evaluation and they do not segregate between CMIP5 and CMIP6.

Line 382: Z20 do not say this, instead they argue that this is a possibility that should be investigated.

References:
Elsaesser, G.S., C.W. O'Dell, M.D. Lebsock, R. Bennartz, and T.J. Greenwald, 2017: The Multi-Sensor Advanced Climatology of Liquid Water Path (MAC-LWP). J. Climate, 30, no. 24, 10193-10210, doi:10.1175/JCLI-D-16-0902.1.

Lebsock, M., and Su, H. (2014),  Application of active spaceborne remote sensing for understanding biases between passive cloud water path retrievals, J. Geophys. Res. Atmos., 119,  8962–8979, doi:10.1002/2014JD021568.

Li, J.-L. F., Forbes, R. M., Waliser, D. E., Stephens, G., and Lee, S. (2014),  Characterizing the radiative impacts of precipitating snow in the ECMWF Integrated Forecast System global model, J. Geophys. Res. Atmos.,  119,  9626–9637, doi:10.1002/2014JD021450.

---

## Author Response (AR2)

**Dear editor, thank you for giving us the opportunity to address the reviewers' concerns. As recommended by reviewer #2, we deleted all comparisons of model results with observational data for cloud fraction, ice water path and liquid water path from the Section 3.2 (now renamed as "Present-day cloud fields") and Figures 4, 5, 6, and 9. Tables 3 and 4 have been adjusted accordingly. Table 2 and Figure 3 have been removed. In order to compare the differences in present-day cloud properties from the three ECS model groups with each other, we combined the model results (with no observations) from Figures 3, 4 and 5 into a new Figure 3 in Section 3.2.**

Reviewer #2

In response to my concerns, the authors have added caveats to the paper that the model evaluation of cloud properties is qualitative rather than quantitative. However, they largely leave Section 3.2 (Evaluation of Cloud Properties) unchanged. In this section, numerical values of regional averages are compared between models and observations, global mean values are compared between models and observations, root mean square differences between models and observations are computed, and spatial pattern correlations between models and observations are computed, in all cases for the three groups of models. Hence Section 3.2 remains almost entirely a quantitative evaluation of modeled cloud properties against observations.

**We thank Reviewer #2 for helping us to improve the manuscript. We addressed all comments in the revised version and in our point-by-point answers below (given in bold). If not otherwise noted, all line numbers refer to the revised manuscript.**

**As recommended, we removed all comparisons of model results with observational estimates for cloud fraction, liquid water path and ice water path in Section 3.2 and throughout the rest of manuscript including all figures and tables (see answer above).**

The authors acknowledge "This study is therefore not possible with the available CMIP model output generated by satellite simulators." I agree that rigorously evaluating models against observations is not possible across the whole suite of CMIP5 and CMIP6 models as attempted here because most models lack COSP output that makes for a meaningful and reliable comparison. This is why I originally raised this concern, and I remain confused as to why the authors continued down this path. Moreover, I disagree that one can "qualitatively" evaluate models against observations while relying on comparing fundamentally different quantities. This could be misleading at best and simply wrong at worst. Therefore I don't believe the authors can confidently say make statements like "we found that models with a high climate sensitivity typically have a better representation of observed cloud properties than models with a low or medium ECS." I recommend removing Section 3.2 entirely, or focusing only on fields for which model-observations differences in the definition of the field can be minimized.

**We deleted now all comparisons of the cloud properties cloud fraction and ice and liquid water path with satellite observations in Section 3.2 and throughout the rest of the manuscript. Only a comparison of TOA cloud radiative effects with CERES-EBAF remains as this dataset has been "developed to be compared directly with climate model results without the need for simulators or other sampling strategies" (CERES-EBAF Expert Developer Guidance). Our statement on a better representation of cloud properties by high ECS models has been deleted from the abstract and**

strictly restricted to results from the comparison with CERES-EBAF data in Section 3.2. A new figure 3 in Section 3.2 now compares the ECS group mean climatologies for these cloud properties with each other only (no satellite observations) to assess systematic differences among the three model groups in today's climate before looking into projected changes.

My concern about comparing models and observations from different time periods remains, but is secondary to the more fundamental concern above. Arguably the least ambiguous way of comparing models and observations would be to use COSP output from atmosphere-only AMIP simulations in which the observed radiative forcings, SSTs, and sea ice concentrations are prescribed, and choosing the period of perfect temporal overlap between models and observations.

We see the point of the reviewer. Since we are comparing only multi-year climatologies, however, the exact time period as well as whether SSTs and sea ice concentrations are prescribed (AMIP runs) or coupled online to the model (historical runs) have very little influence. This is demonstrated in Figure 1 below showing the AMIP output for the same time period (1985-2004) as the historical runs of most models considered in the paper (AMIP output was not available from all models). The differences between these two climatologies are typically quite small. This confirms findings of previous studies (e.g. Lauer et al., 2023) that in general, the skill of CMIP5 and CMIP6 AMIP multi-model means in reproducing the observed cloud climatologies in terms of global means, biases, pattern correlations and RMSDs does not systematically differ from the ones obtained from the historical simulations. As it is the coupled model configuration used in the historical runs that is used for the climate projections analysed in this paper, we decided to keep the analysis of the historical model runs.

The time period for the models (1985-2004) has been chosen to maximize the overlap of the 20-year periods from different generations of models (CMIP5 and CMIP6). While this choice of model years is somewhat arbitrary, we found that it has very little impact on the multi-year multi-model averages. This is not surprising as ESMs are not expected to reproduce the exact observed phase of climate modes largely controlling present-day variability of clouds but rather their statistical properties.

For clarification, we added the following paragraph to Section 3.2 (l. 196-200):

"Biases in simulated sea surface temperatures (SSTs) can affect simulated cloud properties. We therefore also analyzed some results from AMIP simulations that use the atmosphere components of the CMIP models and for which SSTs and sea ice concentrations from observations are prescribed (not shown). Similar to previous studies (e.g. Lauer and Hamilton, 2013; Lauer et al., 2023), we found rather little differences in the multi-year climatologies of cloud properties from the models between the historical and AMIP runs analyzed here."

[Figure]

*Fig. 1: Geographical map of the multi-year annual mean net cloud radiative effect from (a) CERES EBAF Ed4.2 (OBS) and (b,c,d) group means of AMIP simulations (1985-2004) from all three ECS groups.*

I do not have a problem with Sections 3.1 or 3.3, though I am still not sure we learn much from these sections that are not already well established. If Section 3.2 were removed or trimmed to focus only on fields that can rigorously be compared between models and observations, I could support publication of this paper.

**We significantly shortened Section 3.2 by deleting all comparisons of model results for cloud fraction, liquid water path and ice water path with observational data. We only kept a comparison with TOA radiative fluxes from CERES-EBAF Ed. 4.2, that have been specifically tailored for climate model evaluation as suggested in the first round of reviews.**

**Reviewer #3**

Review of "Cloud properties and their projected changes in CMIP models with

low/medium/high climate sensitivity" by L. Bock and A. Lauer.

In this study, the authors classify outputs of 51 CMIP5 and CMIP6 models into low, medium and high ECS groups and then compare them with observations. They further look at the change in cloud properties between historical and 4xCO2 simulations weighted by global mean surface warming. They find that the models from the high ECS group better represent the observed climatology of cloud-related variables and have different sensitivities to warming than the low- and medium-ECS groups.

The topic of this paper aligns well with the scope of the journal. While I recognize the value of investigating the response of clouds to warming and the amount of work it takes in terms of data processing, I find that the study suffers from two major flaws: a non-consistent direct comparison of model cloud-related outputs with observations and the use of coupled historical simulations in the model evaluation. Also, it seems that the authors have already evaluated these models in a separate paper, so I see little value in doing this again. However, I find the analysis of the cloud response to climate change very interesting. More details are given below.

**We also thank Reviewer #3 for the constructive comments that helped improving the manuscript. We addressed all comments in the revised version and in our point-by-point answers below (given in bold). If not otherwise noted, all line numbers refer to the revised manuscript.**

Main comments:

My biggest concern is the comparison of cloud-related fields with observations, which doesn't account for observational uncertainties and inherent limitations of the satellite instruments. The LWP products suffer from large uncertainties (sometimes several times greater than the observed value itself, Lebsock and Su, 2014; Elsaesser et al., 2017) and cannot be used to assess models on a global scale. IWP products seem to more reliable but there is still the question of whether precipitation is accounted for or not (e.g., Li et al., 2014). The cloud fraction also cannot be compared directly to observations because of the instrument limitations and the difference in cloud definitions between models and observations. I'm attaching a figure showing the impact of using ISCCP (basically AVHRR), MODIS and CALIPSO simulators on the original output of the model for 3 CMIP6 models. The differences are very large, region dependent and model dependent…

**Following the suggestion of reviewer #2, we deleted all comparisons of model results with satellite data for total cloud fraction and ice and liquid water path throughout the manuscript including all figures and tables (see details above). Section 3.2 has been shortened and now focuses on a discussion of the differences in present-day cloud properties between the three ECS model groups.**

The second main concern is the comparison of historical simulations with present-day observations that have not the same surface forcings. The SST pattern and magnitude, which have strong impact on all the variables that are studied here, are not well reproduced by the coupled models as shown in the literature (e.g., Seager et al., 2019). Therefore, it is not a fair comparison. Instead, the authors should use AMIP type simulations to assess the models.

**We agree with the reviewer that biases in simulated sea surface temperatures and sea ice concentrations can (and do) affect simulated cloud properties. Regarding multi-year multi-model annual mean climatologies, however, the differences between historical and AMIP simulations are found to be rather small. The same is true for the exact time period chosen for the models that we found to have very little impact on the multi-year group averages. For details and the extension of the text please see our answer to reviewer #2 and Figure 1.**

Another main comment, which could be easily fixed, is the conclusion. Except for the last paragraph, which is very insightful, the conclusion is far too long (2 pages) and does not summarize the results but rather re-state them without any apparent structure.

**As recommended, we condensed the section summary and conclusions by shortening the summary part considerably.**

Minor comments:

Almost no information about the observations used is given and including potential uncertainties, which are raised here and there in the manuscript but without being formally quantified. As is, it looks like the authors have very little knowledge about the observations they're using.

**In the revised version of the manuscript, the only observational dataset used is from CERES-EBAF. We extended the description of CERES-EBAF in Sect. 2.2 providing more details on the dataset including uncertainty estimates for the net cloud radiative effect used in this study.**

I couldn't find a clear definition of how the feedbacks are computed.

**We extended the description of the cloud feedback calculation in the paper as follows: "The net cloud feedback is defined as change in the sum of shortwave and longwave cloud radiative effects at the top of the atmosphere (TOA) per degree of surface warming (2-m temperature) calculated as the difference between abrupt-4×CO2 simulations and the corresponding piControl simulations. The TOA shortwave and longwave cloud radiative effects are calculated as the differences between the respective TOA all-sky and clear-sky radiative fluxes." (l. 108-111)**

The introduction is not doing the best job at motivating the ECS discrimination. If the idea is that larger cloud feedback could be related to mean state cloud properties, then I would classify the models by the global mean feedback rather than ECS, because the ECS is affected by other feedbacks than those from clouds. The way it is presented in the paper is even slightly backward in my opinion.

**In this study is intended as a contribution to the question whether there are systematic differences among CMIP models with different ECS. As cloud feedbacks are an important contribution to the modeled ECS, we focus on projected changes in cloud properties in the models. Sorting the models by simulated cloud feedbacks would make any conclusions of the differences found for ECS more indirect and more difficult to assess. We made this point clearer by adding the following sentence to the introduction (l. 59-62):**

**"A number of different feedbacks are relevant to ECS with cloud feedbacks being an important contribution. In order to assess whether there are systematic differences in simulated cloud properties among model with different ECS, we compare the simulated cloud properties from three groups of models sorted by their ECS values and quantify how the projected changes in cloud properties and cloud radiative effects differ."**

I question the usefulness of having 2 to 3 versions of a model with the same atmospheric component especially when it comes to evaluating atmospheric quantities. They have disproportional impact on the mean. This question is not specific to that study though.

**We agree that there are several models with a similar or even the same atmospheric component which might skew the multi-model means. To our knowledge, however, there is no established general way of considering model inter-dependence in calculating multi-model means, which is probably the reason why multi-model studies typically do not consider a weighing of individual models. Our study is no exception here.**

L129-130: then why is there no distinction between CMIP5 and CMIP6 models?

**The differences between the ECS groups are typically larger than the differences between CMIP5 and CMIP6. A comparison of cloud properties from CMIP5 and CMIP6 is part of the Lauer et al. (2023) paper and thus not repeated here.**

L136-139: I'm not sure what this means. There can be ice clouds at all latitudes in the high levels. This latitude loosely corresponds to where clouds can be mixed-phase cloud almost year-round. These clouds show different feedbacks from the warm clouds.

**We actually meant the presence of ice in the clouds in present-day climate, for which the cloud phase feedback leads to an overall negative net cloud feedback. We clarified this in the revised text (l. 126-129):**

**"[…] corresponds to the latitude region where a change from clouds with an ice component in the piControl simulations to clouds consisting almost entirely of liquid droplets in the abrupt-4xCO2 experiment (cloud phase feedback) starts to contribute significantly to the total cloud feedback (Ceppi et al., 2017)."**

L158: simulated is written two times in a row and I don't think that Z20 is saying or showing such a conclusion in their study.

**Thanks for spotting this. We deleted one "simulated" and replaced "mean state" with "representation of clouds" and "correlated" with "related" to match exactly the statement of Zelinka et al. (2020): "The representation of cloud properties in ESMs is related to the simulated cloud feedback (Zelinka et al., 2020)." (l. 139).**

Fig. 3: Aside from the non-consistent model-to-observations comparison, the spread and SD appear to be very similar between the group, so the differences are not significant and there is no clear systematic behavior among the groups.

**We removed this figure as we deleted the comparisons with observations for the cloud properties as suggested by reviewer #2.**

L198-199: I don't understand this sentence.

**We removed this sentence as we deleted the comparisons of the model results with observations.**

L243: not the number on the figure, so I suppose the authors decided to switch the default dataset to CERES during the first round of review, but failed to revise the text.

**Thanks for spotting this. This has been corrected in the revised version of the manuscript.**

I don't see the added value of Fig. 10 and 11 compared to Fig. 9. I think that Fig. 9 and the analysis associated to it is the best part of this manuscript and should be the focus of this study.

**We think that the different cloud regimes shown in Figures 10 and 11 do have an additional value compared to the zonal means shown in Figure 9 as different cloud types react differently to warming. This behaviour is easily masked in the zonal means averaging over different cloud types. Additionally, the model spread and the uncertainties help with assessing which differences are significant.**

Each main Sc decks is singled out yet no there is no motivation for doing this, I'm also not a fan of picking fixed regions to study cloud response to warming since these decks can evolve in terms of location.

**As the qualitative differences between the different stratocumulus decks are rather small, we combined the three stratocumulus regions in Figure 11 into a single panel that is now included in the new Figure 7. The simplification of using fixed regions seemed fine to us given that the spread among models in e.g. the stratocumulus cloud cover is rather large and thus the exact region is probably of secondary importance.**

The beginning of the conclusion is confusing. The authors argue that the increase of ECS between CMIP generation motivated them to investigate the response of clouds to climate change, yet most of the study is based on present-day climate evaluation and they do not segregate between CMIP5 and CMIP6.

**The increase in ECS between CMIP generations opened up a highly discussed topic about possible reasons and how realistic this is. This motivated us to look in more detail into the models. But not all CMIP6 models have higher ECS values than the CMIP5 models, there are also CMIP5 models in the high ECS group. As we were interested in whether there are systematic differences in projected cloud properties among the models contributing to differences in ECS, we sorted the CMIP models in three ECS groups. Most of the present-day evaluation with satellite data has been removed (see detailed comments above). The focus is now on projected changes in cloud properties in the models and the differences in present-day fields among the model groups that are a starting point for the following discussion of the projections.**

Line 382: Z20 do not say this, instead they argue that this is a possibility that should be investigated.

**Statement has been removed.**

References:

Elsaesser, G.S., C.W. O'Dell, M.D. Lebsock, R. Bennartz, and T.J. Greenwald, 2017: The Multi-Sensor Advanced Climatology of Liquid Water Path (MAC-LWP). J. Climate, 30, no. 24, 10193-10210, doi:10.1175/JCLI-D-16-0902.1.

Lebsock, M., and Su, H. (2014), Application of active spaceborne remote sensing for understanding biases between passive cloud water path retrievals, J. Geophys. Res. Atmos., 119, 8962–8979, doi:10.1002/2014JD021568.

Li, J.-L. F., Forbes, R. M., Waliser, D. E., Stephens, G., and Lee, S. (2014), Characterizing the radiative impacts of precipitating snow in the ECMWF Integrated Forecast System global model, J. Geophys. Res. Atmos., 119, 9626–9637, doi:10.1002/2014JD021450.

---

## Author Response (AR3)

**Dear editor, thank you for guiding our paper through peer-review. In the following we give a point-by-point reply to all comments from the two reviewers. The original reviewers' comments are given in regular, our answers in bold. All line numbers refer to the revised manuscript.**

Reviewer #2

The authors have substantially revised the paper and I am largely satisfied with the changes. I have a few remaining comments:

L114-116: This does not make sense. It is exactly because coupled models simulate their own distinct transient evolution that a given time period will be characterized by a climate state different from each other and from observations. To do the most apples-to-apples comparison of cloud properties between model groupings and between models and observations, one should use AMIP simulations from the overlapping period. That way, differences are unambiguously attributable to model physics, not the state of the atmosphere (notably, SST pattern differences and aerosol loading differences). I am not asking you to do this, but I think this statement should be rephrased for accuracy.

**The point of the reviewer is well taken. We compare, however, on purpose the historical simulations with the CERES-EBAF dataset as it is this model configuration that is being used for projections of future climate. For this reason, we are interested in the performance of the coupled system, not the atmosphere-only models as relevant errors might also be introduced by biases in the simulated ocean surface properties. Time periods not fully matching are a challenge for this kind of comparison. For multi-model climatologies, however, the exact time period is found to make very little difference. We emphasized this point by adding "when comparing long-term climatologies" to the corresponding sentence (l. 93/94):**

**"[…] that this difference in the time periods has very little impact on the multi-year group averages when comparing multi-year climatologies of cloud parameters."**

L 167: Consider citing Zelinka et al (2022) here, as they recently investigated this connection. https://agupubs.onlinelibrary.wiley.com/doi/full/10.1029/2021JD035198

**Thanks, we added the reference (l. 142).**

L169-171: Here it is shown that high ECS models tend to have higher IWP in the midlatitudes. This is stated to be consistent with the hypothesis that improved representation of supercooled liquid leads to higher ECS. This statement is not right on two levels. Historically GCMs have had too much ice and not enough liquid, so higher IWP would typically not be considered as an improvement. Secondly, models with larger mean-state ice tend to have a larger negative phase feedback as that ice transitions to liquid with warming. In isolation, this would mean high IWP models would tend to have lower ECS.

**We removed this statement.**

L214-215: Is it really necessary to reproduce this statement via direct quote?

**We rephrased the sentence with no direct quote (l. 149/150).**

L343-345: What does this statement have to do with the results shown here? There is no evaluation

of whether the models over- or under-estimate cloud feedback in stratocumulus regions. Moreover, inter-model differences in cloud feedback strength is governed by inter-model differences in representation of cloud physics, not in pattern effects (especially in response to a large warming signal like analyzed here). Suggest deleting this statement.

**As suggested, we deleted this statement.**

Reviewer #3

The authors have taken care of most of my concerns in this new version. Some minor comments remain, but should be fairly easy to address. I'm listing them below

A follow up comment to one of the authors' response:
"The increase in ECS between CMIP generations opened up a highly discussed topic about possible reasons and how realistic this is. This motivated us to look in more detail into the models. But not all CMIP6 models have higher ECS values than the CMIP5 models, there are also CMIP5 models in the high ECS group. As we were interested in whether there are systematic differences in projected cloud properties among the models contributing to differences in ECS, we sorted the CMIP models in three ECS groups. Most of the present-day evaluation with satellite data has been removed (see detailed comments above). The focus is now on projected changes in cloud properties in the models and the differences in present-day fields among the model groups that are a starting point for the following discussion of the projections."
I understand your point of view now that you have elaborated. However, you didn't change the conclusion opening and the idea that you convey in the above statement doesn't really appear in this opening.
The direct link between change in cloud properties with respect to climate change may have contributed to ECS is not done in the conclusion. I would recommend implementing some form of this sentence in it "As we were interested in whether there are systematic differences in projected cloud properties among the models contributing to differences in ECS, we sorted the CMIP models in three ECS groups."

**As suggested, we rephrased parts of the first section "summary and conclusions" as follows (l. 311-313):**

**"Of particular interest was whether there are systematic differences in projected cloud properties among the models contributing to differences in ECS. We therefore sorted 51 CMIP5 and CMIP6 models providing the required output in three equally sized ECS groups."**

Other minor comments (line numbers of the highlighted version):
L55-56: Not sure this sentence still makes sense now that the comparison with observations has been removed.

**We removed the sentence.**

L60: I would remove the comma after both.

**Thanks for spotting this, we deleted the comma.**

Cloud feedback computations: The method used to compute cloud feedbacks works well for the SW component between 60°S/N, but it is not accurate in the polar regions and everywhere for the LW component (Shell et al., 2008). Therefore, it should be clearly mentioned that this cloud feedbacks are not adjusted for the effect of non-cloud feedbacks.

**We added the following two sentences to the description of the cloud feedback calculations (l. 110-114):**

**"While this method is commonly used to calculated cloud feedbacks, we would like to note that the results using this method are not exactly the same as those calculated with a more accurate offline radiative transfer method (Soden et al., 2004). Particularly the shortwave CRE can be affected in regions with high surface albedos such as polar latitudes if the surface albedo changes between the two model simulations e.g. because of melting sea ice (Shell et al., 2008)."**

L264-269: I don't quite follow the logic. The high ECS models produce larger IWP values over the SO and more clouds, therefore it proves that more realistic supercooled clouds lead to less negative cloud phase feedback? If anything, I would think the opposite. Higher amount of ice in the high-ECS model would suggest a potential for more negative cloud phase feedback, since the ice reservoir to be turn into liquid clouds in response to warming is larger than in the other model groups.

**We removed this statement.**

L275-280: Interesting to see that there is no increase in LWP over the Sc decks although there is an increase in their cloud fraction as shown by the authors, which seems to suggest that the clouds of the group are probably less bright (also consistent with Cesana et al., 2023).

**We added the following two sentences (l. 177-180): "In addition to the higher cloud cover in the high ECS group in these regions the clouds seem to be less bright in comparison the two other groups. This indicates an improvement of the representation of stratocumulus clouds in the high ECS group, which is consistent with the findings of Cesana et al. (2023)."**

L308-312: Could you please quantify and share the results in the supplementary?

**Comparing the three group averaged climatologies for the netCRE from the historical experiments with the ones from the AMIP experiments gives the following results:**
**Global mean bias: 0.3 – 0.6 W/m$^2$**
**Global mean RMSE: 2.7 – 3.3 W/m$^2$**
**Pattern correlation: 0.97 – 0.98**
**The differences between the historical and AMIP annual mean netCRE are below 5 W/m$^2$ (about 5-10% relative difference) throughout most of the globe. In the ITCZ, differencescan reach up to about 10 W/m$^2$ (up to 10% W/m$^2$).**

**We added the following sentences to the text (l. 203-206):**

**"For the net cloud radiative effect, differences between the annual mean climatologies from the historical and AMIP simulations are below 5 W m$^{-2}$ throughout most of the globe but differences in the ITCZ and tropical Atlantic can reach up to about 10 W m$^{-2}$. Globally averaged, the mean bias for the three group averages ranges between 0.3 and 0.6 W m$^{-2}$, RMSE between 2.7 and 3.3 W m$^{-2}$ and pattern correlations between 0.97 and 0.98."**

**As the differences between the historical and the AMIP simulations are rather small, we do not think it is really worth putting this figure into a (then to be newly created) supplementary information.**

L450-452: "in the high ECS group", not the low, unless I don't understand Fig. 3.

**Thanks for spotting. We corrected this to "high" (l. 320).**

**References**

Cesana, G. v., Ackerman, A. S., Črnivec, N., Pincus, R., & Chepfer, H. (2023). An observation-based method to assess tropical stratocumulus and shallow cumulus clouds and feedbacks in CMIP6 and CMIP5 models. Environmental Research Communications, 5(4). https://doi.org/10.1088/2515-7620/acc78a

Shell, K. M., Kiehl, J. T., & Shields, C. A. (2008). Using the radiative kernel technique to calculate climate feedbacks in NCAR's Community Atmospheric Model. Journal of Climate, 21(10), 2269–2282. https://doi.org/10.1175/2007JCLI2044.1

Soden, B. J., A. J. Broccoli, and R. S. Hemler, 2004: On the Use of Cloud Forcing to Estimate Cloud Feedback. *J. Climate*, **17**, 3661–3665, https://doi.org/10.1175/1520-0442(2004)017<3661:OTUOCF>2.0.CO;2.